# Fundamental Limits of Game-Theoretic LLM Alignment: Smith Consistency and Preference Matching

## Abstract

Nash Learning from Human Feedback (`NLHF`) is a game-theoretic framework for aligning large language models (LLMs) with human preferences by modeling learning as a two-player zero-sum game. However, using raw preference as the payoff in the game highly limits the potential of the game-theoretic LLM alignment framework. In this paper, we systematically study using what choices of payoff based on the pairwise human preferences can yield desirable alignment properties. We establish necessary and sufficient conditions for Condorcet consistency, diversity through mixed strategies, and Smith consistency. These results provide a theoretical foundation for the robustness of game-theoretic LLM alignment. Further, we show the impossibility of preference matching—i.e., no smooth and learnable mappings of pairwise preferences can guarantee a unique Nash equilibrium that matches a target policy, even under standard assumptions like the Bradley-Terry-Luce (BTL) model. This result highlight the fundamental limitation of game-theoretic LLM alignment.

## 1 Introduction

Large language models (LLMs), such as OpenAI-o3 (OpenAI, 2025) and DeepSeek-R1 (DeepSeek-AI et al., 2025), have demonstrated impressive capabilities across a wide range of domains, including code generation, data analysis, elementary mathematics, and reasoning (Hurst et al., 2024; Anthropic, 2024; Chowdhery et al., 2023; Touvron et al., 2023; Ji et al., 2025). These models are increasingly being used to tackle previously unsolved mathematical problems, drive scientific and algorithmic discoveries, optimize complex code-bases, and support decision-making processes that were once considered unlikely to be automated in the near future (Bubeck et al., 2023; Eloundou et al., 2024; Novikov et al., 2025).

A key factor behind the popularity and effectiveness of LLMs is alignment: the process by which models learn to interact with human users and accommodate diverse human opinions and values by aligning their outputs with human preferences (Christiano et al., 2017). The traditional method for alignment, reinforcement learning from human feedback (`RLHF`) (Ouyang et al., 2022; Casper et al., 2023; Dong et al., 2024), typically begins by training a reward model on preference data collected from human labelers, often using the Bradley-Terry-Luce (BTL) model (Bradley and Terry, 1952; Luce, 2012),

$$\mathcal{P}(y \succ y' \mid x) = \frac{\exp(r(x,y))}{\exp(r(x,y)) + \exp(r(x,y'))}, \tag{1.1}$$

where $r(x,y)$ is the reward function and $\mathcal{P}(y \succ y'|x)$ is pairwise human preference, i.e., the fraction of individuals who prefer $y$ over $y'$ under prompt $x$. In this framework, a higher scalar score assigned

by the reward model to an LLM-generated response indicates a stronger preference by human labelers. The LLM is then fine-tuned through maximizing the reward to produce responses that are more likely to align with these preferences. However, Munos et al. (2024) points out that reward model can not deal with preference with cycles, and proposes an alternative alignment approach called Nash learning from human feedback (NLHF). Unlike the reward-based methods, NLHF directly uses preference data to train a preference model and formulates LLM finetuning as finding Nash equilibrium in a two-player zero-sum game, also known as a von Neumann game (Myerson, 2013). Specifically, for a given prompt $x$, the LLM's policy $\pi$ competes against an opposing policy $\pi'$ in a pairwise preference contest, where the objective is to find a policy that maximizes its worst-case preference score. Formally, NLHF solves the following min-max optimization problem:

$$\max_{\pi} \min_{\pi'} \mathbb{E}_{x \sim \rho} \left[ \mathbb{E}_{y \sim \pi(\cdot|x), y' \sim \pi'(\cdot|x)} \left[ \mathcal{P} \left( y \succ y' \mid x \right) \right] \right],$$

where $\rho$ is a given distribution over prompts. However, Munos et al. (2024) does not demonstrate the advantages of using the preference as the payoff in the game.

Recently, criteria from both social choice theory (Conitzer et al., 2024; Dai and Fleisig, 2024; Mishra, 2023) and principles related to diversity (Xiao et al., 2024; Chakraborty et al., 2024) has been increasingly employed to scrutinize the alignment of LLM with human preference. Notably, RLHF has been shown to fail both social choice theory considerations (Noothigattu et al., 2020; Siththaranjan et al., 2024; Ge et al., 2024; Liu et al., 2025) and diversity considerations (Xiao et al., 2024; Chakraborty et al., 2024). In contrast, NLHF has been proved to enjoy these desirable properties. It is shown in Maura-Rivero et al. (2025); Liu et al. (2025) that NLHF is *Condorcet consistent* (see Definition 3.2), meaning that the method always outputs the Condorcet winning response, a response that beats every other alternative response in pairwise majority comparisons, whenever one exists. Further, under a no-tie assumption (see Assumption 2.1), Liu et al. (2025) shows that NLHF is *Smith consistent* (see Definition 4.1), meaning that the method always output responses from the Smith set, the smallest nonempty set of responses that pairwise dominate all alternatives outside the set. Moreover, Liu et al. (2025) shows that when human preference is diverse, i.e., there does not exist a single response that beat every other alternative, NLHF avoid collapsing to a single response by adopting a *mixed strategy*.

Despite these advantages, there is no reason why we must use raw preference to design the payoff in the game-theoretic alignment approach. Using alternative payoffs in the game-theoretic LLM alignment framework might also lead to desirable alignment. In this work, we systematically investigate *the fundamental limits of the game-theoretic LLM alignment framework* by analyzing how various choices of payoff influence its ability to satisfy key alignment criteria. We consider the following general game-theoretic alignment problem, involving a mapping of the preference denoted by $\Psi$:

$$\max_{\pi} \min_{\pi'} \mathbb{E}_{x \sim \rho} \left[ \mathbb{E}_{y \sim \pi(\cdot|x)} \mathbb{E}_{y' \sim \pi'(\cdot|x)} \left[ \Psi \left( \mathcal{P}(y \succ y' \mid x) \right) \right] \right]. \tag{1.2}$$

The general problem (1.2) encompasses a range of games. When $\Psi(t) = t$ is the identity mapping, the objective in Equation (1.2) is equivalent to the standard NLHF objective. When $\Psi(t) = \log(t/(1-t))$ and the preference is generated by a BTL model, Equation (1.2) recovers the standard RLHF objective. More importantly, the preference model $\mathcal{P}_\theta$ used in practice is an estimation of true human preference, which can be regarded as a noisy mapping of the ground-truth preference $\mathcal{P}$. Allowing $\Psi$ to be stochastic provides a way to account for the uncertainty and noise inherent in estimating human preferences. It is worthy to note that similar formalism has been proposed for non game-theoretic approach in Azar et al. (2024), which uses an non-decreasing mapping to process the preference.

In this paper, we first discuss when the solution to problem (1.2) is Condorcet consistent and Smith consistent in Section 3 and 4 respectively. Our results show that these desirable properties is insensitive to the exact value of the payoff, revealing the robustness of game-theoretic alignment approaches. As a special case, we discover a natural generalization of RLHF objective that satisfy all these desirable properties. Technically, we develop novel proof techniques that can tackle a general non-symmetric game directly, instead of relying crucially on the symmetric nature of NLHF as in Liu et al. (2025).

In addition, we examine the diversity of the solution by investigating whether the model produces a mixed strategy (Liu et al., 2025), and whether its output can satisfy the criterion of *preference matching* (Xiao et al., 2024), meaning that the model output exactly matches a target policy which

fully accounts for the diversity of human preference. Our findings suggest diversity can be ensured by mixed strategies, but exactly matching a target is difficult for any game-theoretic alignment approach. This reveal a fundamental limitation of game-theoretic alignment approaches.

## 1.1 Summary of Contributions

We summarize our contributions as follows:

- We show that Condorcet consistency is insensitive to the exact value of the payoff (Theorem 3.1), revealing the robustness of game-theoretic alignment approaches.

- We show that Smith consistency can be ensured by further maintaining the symmetry of the game (Theorem 4.2). Moreover, Smith consistent methods automatically preserve the diversity in human preferences by adopting mixed strategies (Corollary 4.2).

- We show that preserving the diversity in human preference strictly, in the sense of preference matching, is impossible in general (Theorem 5.1). This reveals a fundamental limitation of game-theoretic alignment approaches.

Assuming $\Psi$ is continuous, Table 1 provides a concise summary of our mathematical results.

Table 1: Summary of our mathematical results: the necessary and sufficient conditions on $\Psi$ to guarantee certain desirable alignment properties.

| | |
|---|---|
| Condorcet consistency | $\Psi(t) \geqslant \Psi(1/2) \, , \forall \, 1/2 \leqslant t \leqslant 1$ and $\Psi(t) < \Psi(1/2) \, , \forall \, 0 \leqslant t < 1/2$ |
| Mixed & Condorcet consistency | $\Psi(t) + \Psi(1-t) \geqslant 2\Psi(1/2) \, , \forall \, 1/2 \leqslant t \leqslant 1$ and $\Psi(t) < \Psi(1/2) \, , \forall \, 0 \leqslant t < 1/2$ |
| Smith consistency | $\Psi(t) + \Psi(1-t) = 2\Psi(1/2) \, , \forall \, 1/2 \leqslant t \leqslant 1$ and $\Psi(t) < \Psi(1/2) \, , \forall \, 0 \leqslant t < 1/2$ |

## 1.2 Related Works

A general mapping $\Psi$ is first introduced in Azar et al. (2024) to facilitate the analysis of traditional non game-theoretic LLM alignment methodologies. Their objective function, called $\Psi$PO, applies a general mapping $\Psi$ to the original human preference. In this way, they treat RLHF and DPO as special cases of $\Psi$PO under BTL model and argue that they are prone to overfitting. To avoid overfitting, they take $\Psi$ to be identity and arrive at a new efficient algorithm called IPO. Our problem (1.2) can be regarded as the analogy of $\Psi$PO in the context of game-theoretic LLM alignment. Another difference is that rather than focusing on statistical properties like overfitting, our focus is on the alignment properties such as Smith consistency and preference matching. Moreover, they restrict $\Psi$ to be an non-decreasing map, while we allow $\Psi$ to be arbitrary, even stochastic.

Condorcet consistency is one of the dominant concept in the theory of voting (Gehrlein, 2006; Balinski and Laraki, 2010), and Smith consistency is its natural generalization (Shoham and Leyton-Brown, 2008; Börgers, 2010). They are not studied in the context of LLM alignment until recently (Maura-Rivero et al., 2025; Liu et al., 2025). In Maura-Rivero et al. (2025), the authors show that NLHF with a selection probability that deals with ties is Condorcet consistent. Under a no-tie assumption, Liu et al. (2025) show that NLHF is Condorcet consistent and Smith consistent, whereas RLHF is not unless the preference satisfies a BTL model. Further, they show that the probability that the preference satisfies a BTL model is vanishing under an impartial culture assumption, highlighting a key advantage of the NLHF framework.

Several recent works also focus on aligning LLMs with the diverse human preference (Chakraborty et al., 2024; Xiao et al., 2024; Liu et al., 2025). In Chakraborty et al. (2024), the authors introduce a mixture model to account for the opinion of minority group and arrive at the MaxMin-RLHF method. In Xiao et al. (2024), the authors introduce the concept of preference matching and develop the PM-RLHF objective to pursue this goal. Liu et al. (2025) demonstrates that the original NLHF yields a mixed strategy when no Condorcet winning response exists, whereas standard RLHF produces a deterministic strategy, highlighting a potential advantage of NLHF in preserving the diversity of human preferences.

## 2 Preliminaries

Consider a general mapping $\Psi : [0, 1] \to \mathbb{R}$. We apply $\Psi$ to the preference and study the max-min problem (1.2) with this generalized payoff. Any solution $\boldsymbol{\pi}$ employed by the first player at the Nash equilibrium,

$$\boldsymbol{\pi} \in \arg\max_{\boldsymbol{\pi}} \min_{\boldsymbol{\pi}'} \mathbb{E}_{x \sim \rho} \left[ \mathbb{E}_{y \sim \boldsymbol{\pi}(\cdot|x)} \mathbb{E}_{y' \sim \boldsymbol{\pi}'(\cdot|x)} \left[ \Psi \left( \mathcal{P}(y \succ y' \mid x) \right) \right] \right], \tag{2.1}$$

is called a Nash solution to the problem (1.2). The Nash solution is the policy which fully aligned LLMs will perform. Note that the set of Nash solutions remain the same after an overall shift of payoff, that is, changing $\Psi$ to $\Psi + C$ for any constant $C$ will not affect the problem. The original NLHF objective (Munos et al., 2024) corresponds to the special case where $\Psi(t) = t$, equivalent to $\Psi(t) = t - 1/2$, and the resulting game is symmetric (Duersch et al., 2012), meaning that the two players are the same. However, for an arbitrary mapping $\Psi$, the game is usually not symmetric, and we only focus on the Nash solution employed by the first player.

Given a prompt $x$, we consider the set of all possible responses generated by the LLM: $\{y_1, \ldots, y_n\}$, where $n$ is the total number of possible responses. Without any loss of generality, we drop the dependence on the prompt $x$ from now on. For any two distinct response $y$ and $y'$, recall that $\mathcal{P}(y \succ y')$ denote the preference of $y$ over $y'$, defined as the expected proportion of individuals who prefer $y$ over $y'$. By definition, human preference satisfies the condition $\mathcal{P}(y \succ y') + \mathcal{P}(y' \succ y) = 1$ and naturally we let $\mathcal{P}(y \succ y) = 1/2$ (Munos et al., 2024). For any distinct pair of responses $y$ and $y'$, we say that $y$ beats $y'$ if $\mathcal{P}(y \succ y') > 1/2$. Additionally, following Liu et al. (2025), we adopt the No-Tie assumption throughout this paper.

**Assumption 2.1** (No-Tie). *For any distinct responses $y$ and $y'$, we assume that $\mathcal{P}(y \succ y') \neq 1/2$.*

This assumption is both minimal and practically reasonable. First, if the number of labelers is odd, it automatically holds. Even in cases where a tie occurs, it can always be resolved through a more precise comparison.

**Notation.** For any set $A$, we denote its cardinality by $|A|$. For any $n \in \mathbb{N}_+$, we define $[n] := \{1, \ldots, n\}$. We use $\delta_{ij} := \mathbb{1}\{i = j\}$ for $1 \leqslant i, j \leqslant n$. We represent high-dimensional vectors using bold symbols. Any policy $\boldsymbol{\pi}$ over the set of possible responses $\{y_1, \ldots, y_n\}$ can be identified with a vector in $\mathbb{R}^n$, where each entry $\pi_i$ corresponds to the probability assigned to $y_i$ for $i \in [n]$. We then define the support of a policy $\boldsymbol{\pi}$ as $\mathrm{supp}(\boldsymbol{\pi}) := \{y_i \mid \pi_i > 0, i \in [n]\}$. We write $\boldsymbol{\pi} > 0$ if $\pi_i > 0$ for all $i \in [n]$, and similarly, $\boldsymbol{\pi} \geqslant 0$ if $\pi_i \geqslant 0$ for all $i \in [n]$.

## 3 Condorcet Consistency

In this section, we examine Condorcet consistency—a desirable property for LLM alignment inspired by social choice theory—within the generalized game-theoretic LLM fine-tuning framework (1.2). We begin by defining the Condorcet winning response and Condorcet consistency. We then present Theorem 3.1, which characterizes the necessary and sufficient conditions on the mapping $\Psi$ to guarantee Condorcet consistency. Next, we examine the conditions under which $\Psi$ preserves human preference diversity when no Condorcet winner exists and introduce Theorem 3.2. Finally, we discuss the continuity assumption underlying Theorem 3.2.

Following Liu et al. (2025), a response that is preferred over all others in pairwise comparisons by the preference model is referred to as the Condorcet winning response.

**Definition 3.1** (Condorcet Winning Response). A response $y^\star$ is called a Condorcet winning response if $\mathcal{P}(y^\star \succ y) > 1/2$ for all $y \neq y^\star$.

It is clear that there can be at most one Condorcet winning response. When such a response exists, a natural requirement for LLM alignment is that this response should be the output. This property is known as Condorcet consistency.

**Definition 3.2** (Condorcet Consistency). Problem (1.2) is Condorcet consistent if when there exists a Condorcet winning response, the Nash solution to (1.2) is unique and corresponds to this Condorcet winning response.

Liu et al. (2025); Maura-Rivero et al. (2025) show that the original `NLHF` objective, which corresponds to the case where $\Psi(\cdot)$ is identity, is Condorcet consistent. In this paper, we proceed further and investigate the following question:

Which choices of $\Psi$ ensure Condorcet consistency?

We answer this question in Theorem 3.1. The proof is provided in Appendix A.

**Theorem 3.1.** *Problem* (1.2) *is Condorcet consistent if and only if $\Psi(\cdot)$ satisfies*

$$\begin{cases} \Psi(t) \geqslant \Psi(1/2), 1 \geqslant t \geqslant 1/2 \\ \Psi(t) < \Psi(1/2), 1/2 > t \geqslant 0 \end{cases} . \tag{3.1}$$

Note that this condition is much weaker than requiring $\Psi$ to be increasing. It only demands that $\Psi$ maps any value greater than 1/2 to some value larger than $\Psi(1/2)$, and any value less than 1/2 to some value smaller than $\Psi(1/2)$. This implies that a wide range of mapping functions can be used within the game-theoretic LLM alignment framework (1.2) to ensure Condorcet consistency. Furthermore, in practice, we do not have access to the ground-truth preference model. Instead, we parameterize the preference model using a deep neural network, $\mathcal{P}_\theta(y \succ y')$, trained on large-scale preference datasets (Munos et al., 2024). Due to the limitations of the datasets and the optimization process, the learned model only approximates the true human preferences. We can view this approximation as $\Psi(\mathcal{P}(y \succ y'))$ in our framework. In practice, we can enforce the parameterized preference model to satisfy $\mathcal{P}_\theta(y \succ y) = 1/2$, then our results show that as long as this approximation yields the correct pairwise majority comparisons—specifically, that $\mathcal{P}_\theta(y \succ y') \geqslant 1/2 > \mathcal{P}_\theta(y' \succ y)$ whenever $y$ beats $y'$—then the LLM alignment remains Condorcet consistent. This strongly highlights the robustness of the game-theoretic LLM alignment approach in achieving Condorcet consistency.

When a Condorcet winning response does not exist, human preferences are diverse and there is no single response that is better than others. Therefore, in order to preserve the diversity inherent in human preferences, it is natural to require the Nash solution not to collapse to a single response. This motivation leads to the following characterization of diversity through mixed strategies.

**Definition 3.3** (Mixed Strategies)**.** A Nash solution $\pi$ is called a mixed strategy if $|\operatorname{supp}(\pi)| > 1$.

Liu et al. (2025) demonstrates that the original `NLHF`, which corresponds to the case where $\Psi(\cdot)$ is identity, yields a mixed strategy when no Condorcet winning response exists. Assuming that problem (1.2) is Condorcet consistent, we proceed further and investigate:

Which choices of $\Psi$ lead to a mixed strategy in the absence of a Condorcet winning response?

We now focus on mappings $\Psi$ that are continuous at 1/2, a condition commonly encountered in practical learning setups. Under this mild assumption, we answer this question in Theorem 3.2 and the proof is provided in Appendix C.

**Theorem 3.2.** *Assume that the mapping $\Psi(\cdot)$ is continuous at $1/2$. Assuming the Condorcet consistency of problem* (1.2)*, then any Nash solution is mixed when there is no Condorcet winning response if and only if $\Psi(\cdot)$ satisfies*

$$\Psi(t) + \Psi(1-t) \geqslant 2\Psi(1/2), \forall 0 \leqslant t \leqslant 1 \text{ and } \Psi(t) < \Psi(1/2), \forall 0 \leqslant t < 1/2. \tag{3.2}$$

The first condition arises from the requirement of mixed strategies, while the second condition is a reduction of the condition inherited from Theorem 3.1 under the assumption of Condorcet consistency and the first condition.

Choices of payoff functions are harder to characterize when we relax the continuity assumption. The following example investigate a special piece-wise constant mapping, which does not satisfy the first condition in Theorem 3.2.

**Example 3.4.** *Let $M_- < \Psi(1/2) \leqslant M_+$ and take*

$$\Psi(t) = \begin{cases} M_-, & 0 \leqslant t < 1/2 \\ \Psi(1/2), & t = 1/2 \\ M_+, & 1/2 < t \leqslant 1 \end{cases} .$$

*Then, any Nash solution is mixed when there is no Condorcet winning response.*

The proof of Example 3.4 is deferred to Appendix B. This example implies that choices of payoff functions are considerably richer when we relax the continuity assumption.

## 4 Smith Consistency

In this section, we extend the discussion of Condorcet consistency to Smith consistency. First, we define the Smith set and Smith consistency. Next, we present Theorem 4.2, which provides the necessary and sufficient condition for the mapping $\Psi$ to ensure Smith consistency. Finally, we highlight that Smith-consistent methods inherently preserve the diversity present in human preferences and discuss the continuity assumption in Theorem 4.2.

Condorcet consistency only ensures the method capture the right response when there exists a Condorcet winning response. In general, when there is no Condorcet winning response, we can expect that there might be a set of responses satisfying similar property, generalizing Definition 3.1. Under Assumption 2.1, Liu et al. (2025) revealed a more detailed decomposition of the preference structure. Specifically, the set of responses can be partitioned into distinct groups $S_1, \ldots, S_k$, where every response in $S_i$ is preferred over all responses in $S_j$ for $i < j$, summarized in the following theorem.

**Theorem 4.1** (Liu et al. (2025)). *Under Assumption 2.1, the set of responses can be partitioned into disjoint subsets $S_1, \ldots, S_k$ such that:*

    *1. Each $S_i$ either forms a Condorcet cycle or is a single response.*

    *2. For any $j > i$, any response $y \in S_i$ and $y' \in S_j$, $\mathcal{P}(y \succ y') > \frac{1}{2}$.*

*Moreover, this decomposition is unique.*

When $|S_1| = 1$, the response in $S_1$ is exactly the Condorcet winning response. Thus, $S_1$ is the generalization of Condorcet winning response, and is referred as the Condorcet winning set in Liu et al. (2025). Traditionally, a subset with such property is also known as the Smith set in the literature of social choice theory (Shoham and Leyton-Brown, 2008). Here we choose to adopt the name Smith set to distinguish with the concept of Condorcet winning response. Given this decomposition, it is natural to desire that an aligned LLM adopts a strategy supported exclusively on the top group $S_1$, as any response outside $S_1$ is strictly less preferred than any response inside $S_1$. This desirable property is referred to as Smith consistency:

**Definition 4.1** (Smith Consistency). Problem (1.2) is Smith consistent if the support of any Nash solution is contained in the Smith set $S_1$.

Liu et al. (2025) showed that the original NLHF payoff, which corresponds to the case where $\Psi(t) = t$, is Smith consistent. Here, we investigate this question for a general mapping $\Psi$:

$$\textit{Which choices of } \Psi \textit{ ensure Smith consistency?}$$

Here, similar to Theorem 3.2, we answer this question in Theorem 4.2 for mappings that is continuous at $1/2$. The proof is provided in Appendix E.

**Theorem 4.2.** *Suppose that the mapping $\Psi(\cdot)$ is continuous at $1/2$, problem (1.2) is Smith consistent if and only if $\Psi(\cdot)$ satisfies*

$$\Psi(t) + \Psi(1-t) = 2\Psi(1/2), \forall t \in [0,1] \text{ and } \Psi(t) < \Psi(1/2), \forall 0 \leqslant t < 1/2.$$

The first condition $\Psi(t) + \Psi(1-t) = 2\Psi(1/2)$ says nothing but the zero-sum game formed by problem (1.2) is equivalent to a symmetric two-player zero-sum game[1] (Duersch et al., 2012). By definition, Smith consistency implies Condorcet consistency because when there is a Condorcet winning response, $S_1$ is exactly the set whose only element is the Condorcet winning response. Thus, the second condition is just a reduction of the condition in Theorem 3.1 under the first condition. It is easy to see $\Psi(t) = t$ satisfies these conditions, and thus our result generalize Theorem 3.6 in Liu et al. (2025). More interestingly, $\Psi(t) = \log(t/(1-t))$ also satisfies these conditions. This implies that

$$\max_{\boldsymbol{\pi}} \min_{\boldsymbol{\pi}'} \mathbb{E}_{x \sim \rho} \left[ \mathbb{E}_{y \sim \boldsymbol{\pi}(\cdot|x)} \mathbb{E}_{y' \sim \boldsymbol{\pi}'(\cdot|x)} \left[ \log \left( \frac{\mathcal{P}(y \succ y' \mid x)}{\mathcal{P}(y' \succ y \mid x)} \right) \right] \right],$$

which is a natural generalization of standard RLHF when human preferences does not satisfy BTL model, is also Smith consistent.

---

[1]This can be seen by shifting the payoff by $\Psi(1/2)$, which leaves the Nash solution unchanged.

The set of choices for $\Psi$ that ensure Smith consistency is quite broad. We can easily construct such a $\Psi$ by first defining $\Psi(t)$ on $[0, 1/2]$ to satisfy $\Psi(t) < \Psi(1/2)$ for all $t \in [0, 1/2)$, and then extending it to $[0, 1]$ by setting $\Psi(t) = 2\Psi(1/2) - \Psi(1 - t)$ for all $t \in (1/2, 1]$. Moreover, as discussed in Section 3, a practical preference model $\mathcal{P}_\theta(y \succ y')$ can be seen as a mapping of the ground truth preference via $\Psi$, i.e., $\Psi(\mathcal{P}(y \succ y'))$. Thus, the first condition in Theorem 4.2 requires the preference model to satisfy $\mathcal{P}_\theta(y \succ y') + \mathcal{P}_\theta(y' \succ y) = 1$, with $\mathcal{P}_\theta(y \succ y) = 1/2$ enforced. However, several practically used preference models (Munos et al., 2024; Jiang et al., 2023; Wu et al., 2024) do not guarantee this condition, which may cause the aligned LLM strategy to fail to satisfy Smith consistency.

As any mapping satisfying the condition in Theorem 4.2 also satisfies the condition in Theorem 3.2, we obtain the following corollary:

**Corollary 4.2.** *Suppose that the mapping $\Psi(\cdot)$ is continuous at $1/2$. Then if problem (1.2) is Smith consistent, any Nash solution is also mixed.*

This shows that when $|S_1| > 1$, the Nash solution to problem (1.2) with any $\Psi$ such that Smith consistency holds will not only support on $S_1$ but also be a mixed strategy on $S_1$ without collapsing to a single response. As a conclusion, a Smith consistent method can preserve the diversity inherent in human preferences, at least partially.

Lastly, we discuss what happens if $\Psi$ is not continuous at $1/2$. Choices of mappings $\Psi$ are considerably richer and consequently harder to characterize when we relax the continuity assumption. The following example shows that the piece-wise constant mapping in Example 3.4 also ensures Smith consistency.

**Example 4.3.** *Let $M_- < \Psi(\frac{1}{2}) < M_+$, and we take*

$$\Psi(t) = \begin{cases} M_- & 0 \leqslant t < 1/2 \\ \Psi(1/2) & t = 1/2 \\ M_+ & 1/2 < t \leqslant 1 \end{cases}.$$

*Then problem (1.2) is Smith consistent. The proof is provided in Appendix D.*

# 5 Impossibility of Preference Matching

In this section, we first revisit the definition of preference matching in the BTL model (Xiao et al., 2024). Then we introduce a general theoretical framework of preference matching within the context of game-theoretic LLM alignment, and establish a general impossibility result, as stated in Theorem 5.1. Finally we apply this general result to problem (1.2), concluding that preference matching is impossible.

In previous sections, we have characterized the diversity of alignment result via mixed strategies. In Xiao et al. (2024), the authors propose a more refined criterion for diversity when the preference $\mathcal{P}(y \succ y' \mid x)$ follows a BTL model,

$$\mathcal{P}(y \succ y' \mid x) = \frac{\exp(r(x, y))}{\exp(r(x, y)) + \exp(r(x, y'))},$$

as given by (1.1). They point out that it is unwise to completely disregard any minority opinions in the case that 51% of human labelers prefer $y_1$ over $y_2$ for a binary comparison. They suggest that the policy (5.1),

$$\boldsymbol{\pi}^*(y \mid x) = \frac{\exp\left(r(x, y)\right)}{\sum_{y'} \exp\left(r(x, y')\right)}, \tag{5.1}$$

referred to as the preference-matching policy, fully accounts for the diversity in human preferences.

It is easy to see that there exists a Condorcet winning response under BTL model. According to Theorem 3.1, using preference $\Psi(\mathcal{P}(y \succ y' \mid x))$ as payoff with $\Psi(t) = t$ or $\Psi(t) = \log(t/(1-t))$ will lead the Nash solution to collapse to a single response instead of matching with $\boldsymbol{\pi}^*$. This shows that both RLHF and NLHF do not accounts for the diversity inherent in human preferences from the perspective of preference matching (Xiao et al., 2024; Liu et al., 2025), even under BTL model.

To achieve alignment fully accounting for diversity, we would like to match the Nash solution with the desired policy $\boldsymbol{\pi}^*$. In Xiao et al. (2024), the authors answered this question for RLHF. They proposed

the preference matching RLHF (PM-RLHF) method which successfully achieves preference matching, by slightly modifying the RLHF objective. Here, we aim to explore the possibility of designing a new learnable payoff matrix that aligns with the desired strategy in a game-theoretic framework for LLM alignment:

*Which choices of $\Psi$ ensure preference matching?*

Although it is currently unknown how to generalize the notion of preference matching policy to a general non-BTL preference, to maintain the generality of the discussion and drop the BTL model assumption, we suppose there exists an ideal policy, denoted by $\boldsymbol{\pi}^*$, which captures the diversity of human preferences perfectly.

Given a prompt $x$, we consider the set of all possible responses generated by the LLM: $\{y_1, \ldots, y_n\}$. We further suppose that the policy $\boldsymbol{\pi}^*$ has full support over these $n$ responses, meaning $\boldsymbol{\pi}^* > 0$, as we exclude responses not supported by $\boldsymbol{\pi}^*$ from consideration. Then our goal is to construct a game, represented by a payoff matrix $\{\alpha_{ij}\}_{i,j=1}^n$, with its Nash solution the given policy $\boldsymbol{\pi}^*$, i.e.,

$$\boldsymbol{\pi}^* = \arg\max_{\boldsymbol{\pi}} \min_{\boldsymbol{\pi}'} \sum_{i=1}^n \sum_{j=1}^n \alpha_{ij} \pi_i \pi_j'.$$

To answer this question, we characterize the Nash solution under the given payoff matrix $\{\alpha_{ij}\}_{i,j=1}^n$, which is summarized by the following KKT condition. The proof is deferred to Appendix F.1.

**Lemma 5.1** (KKT Condition). *Consider a game with payoff matrix $\{\alpha_{ij}\}_{i,j=1}^n$. Then $\boldsymbol{\pi}^* > 0$ is a Nash solution to the game if and only if there exists $\boldsymbol{u}^* \in \mathbb{R}^n$ with $\boldsymbol{u}^* \geqslant 0$ and $\sum_{i=1}^n u_i^* = 1$, and $t^* \in \mathbb{R}$ such that the following KKT conditions hold:*

$$\begin{cases} \sum_{i=1}^n \pi_i^* \alpha_{ij} - t^* \leqslant 0 & j = 1, \cdots, n \\ u_j^* \left( \sum_{i=1}^n \pi_i^* \alpha_{ij} - t^* \right) = 0 & j = 1, \cdots, n \\ \sum_{j=1}^n \alpha_{ij} u_j^* = t^* & i = 1, \cdots, n \end{cases}.$$

According to Lemma 5.1, it is easy to verify that the payoff matrix

$$\alpha_{ij} = \pi_i^* + \pi_j^* - \delta_{ij}, \forall 1 \leqslant i, j \leqslant n, \tag{5.2}$$

and the payoff matrix

$$\alpha_{ij} = -\frac{\pi_j^*}{\pi_i^*} + n\delta_{ij}, \forall 1 \leqslant i, j \leqslant n, \tag{5.3}$$

both guarantee that $\boldsymbol{\pi}^*$ is a Nash solution (the details are provided in Appendix F.2). However, these payoff matrices do not depend on the given policy $\boldsymbol{\pi}^*$ in a reasonable way. The payoff matrix in Equation (5.2) is symmetric, making it difficult to interpret. Even worse, it depends on the raw value of $\boldsymbol{\pi}^*$. In practice, $\boldsymbol{\pi}^*$ is often only known up to a normalizing constant. For instance, the preference matching policy (5.1) includes a normalizing constant in the denominator that involves summing over $n$ terms. This constant is hard to determine when $n$ is large and unknown, as is often the case in LLMs. The payoff matrix in Equation (5.3) faces a similar issue as it explicitly depends on $n$, which is an extremely large and unknown value in practice.

In summary, the above two payoff matrices rely on information that is often unavailable in practice, such as $n$ and the raw value of $\boldsymbol{\pi}^*$. What we can obtain in practice for the design of $\alpha_{ij}$ is the preference information between two responses $y_i$ and $y_j$, which we assume depends solely on the ratio between $\pi_i^*$ and $\pi_j^*$. When the preference satisfies the BTL model (1.1), this assumption is justified by the fact that the preference between any two responses depends solely on the ratio of the values assigned by their corresponding preference matching policies (5.1). From this practical consideration, we assume that the payoff matrix satisfies the following assumptions:

**Assumption 5.2.** *Given any $\boldsymbol{\pi}^* > 0$, the payoff matrix $\{\alpha_{ij}\}_{i,j=1}^n$ satisfies the following conditions:*

*1. For all $i \in [n]$, $\alpha_{ii} = C$ where $C$ is a constant independent of $\boldsymbol{\pi}^*$ and $n$. In other words, the diagonal elements are the same constant.*

2. For all $i, j \in [n]$ with $i \neq j$, $\alpha_{ij} = f\left(\frac{\pi_i^*}{\pi_j^*}\right)$ for some smooth function $f$ that is independent of $\pi^*$ and $n$. In other words, the off-diagonal elements depend on the ratio $\frac{\pi_i^*}{\pi_j^*}$ in the same way for all pairs $(i, j)$ with $i \neq j$.

We emphasize that the above two assumptions are crucial for constructing a meaningful and practically learnable payoff matrix. Furthermore, for effective alignment, the payoff matrix should not only ensure $\pi^*$ to be a Nash solution, but $\pi^*$ must be the only Nash solution. The uniqueness requirement excludes trivial payoff matrices such as $\alpha_{ij} = C$, where every $\pi^* > 0$ is a Nash solution.

Unfortunately, in Theorem 5.1, we prove that such a payoff matrix $\{\alpha_{ij}\}_{i,j=1}^n$ does not exist generally. The proof can be found in Appendix F.3.

**Theorem 5.1** (Impossibility of Preference Matching for General Payoffs). *There does not exist a payoff matrix $\{\alpha_{ij}\}_{i,j=1}^n$ satisfying Assumption 5.2 such that for any given $\pi^* > 0$, the Nash solution to the game is unique and equals to $\pi^*$.*

**Remark 5.3.** If we relax Assumption 5.2 and allow the entries of the payoff matrix to depend on $n$, then the design (5.3) is actually eligible for preference matching.

Theorem 5.1 implies that no simple mapping of the preference can yield a payoff that leads to preference matching. As a special case of Theorem 5.1, under BTL model, the generalized game in Equation (1.2) with a smooth mapping $\Psi$,

$$\max_{\pi} \min_{\pi'} \mathbb{E}_{x \sim \rho} \left[ \mathbb{E}_{y \sim \pi(\cdot|x)} \mathbb{E}_{y' \sim \pi'(\cdot|x)} \left[ \Psi \left( \mathcal{P}(y \succ y' \mid x) \right) \right] \right] ,$$

cannot achieve preference matching. Therefore, we obtain the following corollary.

**Corollary 5.4.** *Problem (1.2) with smooth mapping $\Psi$ cannot achieve preference matching.*

# 6 Conclusion

We investigate several properties motivated by social choice theory and diversity considerations within the general game-theoretic LLM alignment framework (1.2), where the payoff is designed as a mapping $\Psi$ of the original preference. We identify the necessary and sufficient conditions on $\Psi$ to guarantee Condorcet consistency and Smith consistency. These conditions allow for a considerably broad class of choices for $\Psi$, demonstrating that these desirable alignment properties are not sensitive to the exact values of the payoff, thereby providing a theoretical foundation for the robustness of the game-theoretic LLM alignment approach. Additionally, we examine conditions on $\Psi$ that ensure the resulting policy is a mixed strategy, preserving diversity in human preferences. Finally, we prove that achieving exact preference matching is impossible under the general game-theoretic alignment framework with a smooth mapping, revealing fundamental limitations of this approach.

**Limitations and Discussion.** Our findings suggest several promising directions for future research on LLM alignment. First, while we establish an impossibility result for preference matching under the assumption that $\Psi$ is smooth, it remains an open question whether preference matching can be achieved when $\Psi$ is merely continuous. Second, in practical settings, regularization terms based on the reference model are often added to problem (1.2). Regularization may be crucial for preference matching, for example, Xiao et al. (2024) modify the regularization term in RLHF to achieve preference matching. Analyzing the alignment properties of game-theoretic methods with such regularization is another interesting avenue for future work. Furthermore, how to explicitly define a preference-matching policy for general preferences that do not satisfy the BTL model, and how to develop alignment approaches capable of learning such a policy, remain open problems. Finally, our results highlight that practical preference models must satisfy certain anti-symmetry conditions to ensure Smith consistency — conditions that are not guaranteed by several currently used models. Thus, designing preference model architectures that enforce anti-symmetry is an important and interesting future direction.

**Broader Impacts.** The goal of this paper is to investigate several theoretical properties of the general game-theoretic LLM alignment approach. There are many potential societal consequences of our work, none of which we feel must be specifically highlighted here.

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

## A   Proof of Theorem 3.1

**Notation.**   For simplicity, we denote $\Psi(\mathcal{P}(y_i \succ y_j))$ as $\Psi_{ij}$ for any $1 \leqslant i, j \leqslant n$, and define the payoff matrix as $\boldsymbol{\Psi} := \Psi_{ij \, 1 \leqslant i,j \leqslant n}$. We then define the total payoff by:

$$\mathcal{P}_{\Psi}(\boldsymbol{\pi}_1, \boldsymbol{\pi}_2) := \sum_{i=1}^{n} \sum_{j=1}^{n} \pi_{1,i} \pi_{2,j} \Psi_{ij} \, .$$

We denote by $\boldsymbol{\delta}_i$ the policy supported solely on $y_i$, i.e., $\mathrm{supp}(\boldsymbol{\delta}_i) = \{y_i\}$. The mixed policy $(\boldsymbol{\delta}_{i_1} + \ldots + \boldsymbol{\delta}_{i_k})/k$ is then defined as the policy $\boldsymbol{\pi}$ such that

$$\pi_i = \begin{cases} 1/k, & i \in \{i_1, \cdots, i_k\} \\ 0, & \text{otherwise} \end{cases} ,$$

for any subset $\{i_1, \ldots, i_k\} \subseteq [n]$.

*Proof of Theorem 3.1.*  Without any loss of generality, we assume that $y_1$ is the Condorcet winning response. First, we show that a necessary condition that ensures the Condorcet consistency of problem (1.2) is:

$$\begin{cases} \Psi(t) \geqslant \Psi(1/2) \, , 1 \geqslant t \geqslant 1/2 \\ \Psi(t) < \Psi(1/2) \, , 0 \leqslant t < 1/2 \end{cases} . \tag{A.1}$$

To show this, we examine the case where $n = 2$. For any $1 \geqslant t > 1/2$, we consider the game with the payoff in Table 2. By the definition of Condorcet consistency, all Nash equilibrium of this game is of the form $(\boldsymbol{\delta}_1, \boldsymbol{\pi}^\star)$ for some $\boldsymbol{\pi}^\star$.

Table 2: Payoff matrix with two responses $\{y_1, y_2\}$.

| $\Psi(\mathcal{P}(y \succ y'))$ | $y' = y_1$ | $y' = y_2$ |
|:---:|:---:|:---:|
| $y = y_1$ | $\Psi(1/2)$ | $\Psi(t)$ |
| $y = y_2$ | $\Psi(1-t)$ | $\Psi(1/2)$ |

**Case 1.**   If $\Psi(t) > \Psi(1/2)$, we have

$$\boldsymbol{\pi}^\star = \arg \min_{\boldsymbol{\pi}} \mathcal{P}_{\Psi}(\boldsymbol{\delta}_1, \boldsymbol{\pi}) = \arg \min_{\boldsymbol{\pi}} \{\pi_1 \Psi(1/2) + \pi_2 \Psi(t)\} = \boldsymbol{\delta}_1 \, .$$

Therefore, we have

$$\Psi(1/2) = \mathcal{P}_{\Psi}(\boldsymbol{\delta}_1, \boldsymbol{\delta}_1) = \max_{\boldsymbol{\pi}} \mathcal{P}_{\Psi}(\boldsymbol{\pi}, \boldsymbol{\delta}_1) \geqslant \mathcal{P}_{\Psi}(\boldsymbol{\delta}_2, \boldsymbol{\delta}_1) = \Psi_{21} = \Psi(1-t) \, ,$$
$$\Psi(1/2) = \mathcal{P}_{\Psi}(\boldsymbol{\delta}_1, \boldsymbol{\delta}_1) = \min_{\boldsymbol{\pi}} \mathcal{P}_{\Psi}(\boldsymbol{\delta}_1, \boldsymbol{\pi}) \leqslant \mathcal{P}_{\Psi}(\boldsymbol{\delta}_1, \boldsymbol{\delta}_2) = \Psi_{12} = \Psi(t) \, .$$

Hence, we have $\Psi(1-t) \leqslant \Psi(1/2) < \Psi(t)$. If $\Psi(1/2) = \Psi(1-t)$, notice that

$$\mathcal{P}_{\Psi}(\boldsymbol{\pi}, \boldsymbol{\delta}_1) = \pi_1 \Psi(1/2) + \pi_2 \Psi(1-t) = \Psi(1/2) \implies \boldsymbol{\delta}_2 \in \arg \max_{\boldsymbol{\pi}} \mathcal{P}_{\Psi}(\boldsymbol{\pi}, \boldsymbol{\delta}_1) \, ,$$
$$\mathcal{P}_{\Psi}(\boldsymbol{\delta}_2, \boldsymbol{\pi}) = \pi_1 \Psi(1-t) + \pi_2 \Psi(1/2) = \Psi(1/2) \implies \boldsymbol{\delta}_1 \in \arg \min_{\boldsymbol{\pi}} \mathcal{P}_{\Psi}(\boldsymbol{\delta}_2, \boldsymbol{\pi}) \, .$$

Therefore, $(\boldsymbol{\delta}_2, \boldsymbol{\delta}_1)$ is also a Nash equilibrium, which causes a contradiction to the fact that problem (1.2) is Condorcet consistent. Therefore, we have $\Psi(t) > \Psi(1/2) > \Psi(1-t)$ for any $1 \geqslant t > 1/2$.

**Case 2.**   If $\Psi(t) < \Psi(1/2)$, we have

$$\boldsymbol{\pi}^\star = \arg \min_{\boldsymbol{\pi}} \mathcal{P}_{\Psi}(\boldsymbol{\delta}_1, \boldsymbol{\pi}) = \arg \min_{\boldsymbol{\pi}} \{\pi_1 \Psi(1/2) + \pi_2 \Psi(t)\} = \boldsymbol{\delta}_2 \, .$$

However, notice that

$$\Psi(1/2) = \mathcal{P}_{\Psi}(\boldsymbol{\delta}_2, \boldsymbol{\delta}_2) \leqslant \max_{\boldsymbol{\pi}} \mathcal{P}_{\Psi}(\boldsymbol{\pi}, \boldsymbol{\delta}_2) = \mathcal{P}_{\Psi}(\boldsymbol{\delta}_1, \boldsymbol{\delta}_2) = \Psi(t) < \Psi(1/2) \, ,$$

which causes a contradiction.

**Case 3.** If $\Psi(t) = \Psi(1/2)$. When $\Psi(1-t) = \Psi(1/2)$, any $(\boldsymbol{\pi}_1, \boldsymbol{\pi}_2)$ is a Nash equilibrium, which causes a contradiction to the fact that problem (1.2) is Condorcet consistent. When $\Psi(1-t) > \Psi(1/2)$, note that

$$\mathcal{P}_\Psi(\boldsymbol{\delta}_2, \boldsymbol{\pi}) = \pi_1 \Psi(1-t) + \pi_2 \Psi(1/2) \geqslant \Psi(1/2) \Longrightarrow \boldsymbol{\delta}_2 \in \arg\min_{\boldsymbol{\pi}} \mathcal{P}_\Psi(\boldsymbol{\delta}_2, \boldsymbol{\pi}),$$

$$\mathcal{P}_\Psi(\boldsymbol{\pi}, \boldsymbol{\delta}_2) = \pi_1 \Psi(t) + \pi_2 \Psi(1/2) = \Psi(1/2) \Longrightarrow \boldsymbol{\delta}_2 \in \arg\max_{\boldsymbol{\pi}} \mathcal{P}_\Psi(\boldsymbol{\pi}, \boldsymbol{\delta}_2).$$

Therefore, $(\boldsymbol{\delta}_2, \boldsymbol{\delta}_2)$ is a Nash equilibrium, which also causes a contradiction to the fact problem (1.2) is Condorcet consistent. Hence, we have $\Psi(1-t) < \Psi(1/2)$.

In summary, for any $1 \geqslant t > 1/2$, we have $\Psi(1-t) < \Psi(1/2) \leqslant \Psi(t)$. Hence, (A.1) holds if problem (1.2) is Condorcet consistent. Next, we prove that (A.1) is also sufficient for the Condorcet consistency of problem (1.2). Recall that $\Psi_{i1} = \Psi(\mathcal{P}(y_i \succ y_1)) < \Psi(1/2)$, and $\Psi_{1i} = \Psi(\mathcal{P}(y_1 \succ y_i)) \geqslant \Psi(1/2)$ for any $i \neq 1$. If $(\boldsymbol{\pi}_1^\star, \boldsymbol{\pi}_2^\star)$ is a Nash equilibrium. Notice that

$$\mathcal{P}_\Psi(\boldsymbol{\pi}_1^\star, \boldsymbol{\pi}_2^\star) = \max_{\boldsymbol{\pi}} \mathcal{P}_\Psi(\boldsymbol{\pi}, \boldsymbol{\pi}_2^\star) \geqslant \mathcal{P}_\Psi(\boldsymbol{\delta}_1, \boldsymbol{\pi}_2^\star) = \sum_{i=1}^n \pi_{2,i}^\star \Psi_{1i},$$

$$\mathcal{P}_\Psi(\boldsymbol{\pi}_1^\star, \boldsymbol{\pi}_2^\star) = \min_{\boldsymbol{\pi}} \mathcal{P}_\Psi(\boldsymbol{\pi}_1^\star, \boldsymbol{\pi}) \leqslant \mathcal{P}_\Psi(\boldsymbol{\pi}_1^\star, \boldsymbol{\delta}_1) = \sum_{i=1}^n \pi_{1,i}^\star \Psi_{i1}.$$

Therefore, if $\boldsymbol{\pi}_1^\star \neq \boldsymbol{\delta}_1$, we have

$$\Psi(1/2) \leqslant \sum_{i=1}^n \pi_{2,i}^\star \Psi_{1i} \leqslant \mathcal{P}_\Psi(\boldsymbol{\pi}_1^\star, \boldsymbol{\pi}_2^\star) \leqslant \sum_{i=1}^n \pi_{1,i}^\star \Psi_{i1} < \Psi(1/2), \tag{A.2}$$

which causes a contradiction. Therefore, $\boldsymbol{\pi}_1^\star = \boldsymbol{\delta}_1$, i.e., problem (1.2) is Condorcet consistent. Hence, we conclude our proof. $\qquad\square$

# B  Proof of Example 3.4

*Proof of Example 3.4.* We prove this conclusion by contradiction. Suppose that the Nash solution is $\boldsymbol{\delta}_{i^\star}$ for some $i^\star \in [n]$, and the Nash equilibrium is $(\boldsymbol{\delta}_{i^\star}, \boldsymbol{\pi}^\star)$. As there is no Condorcet winning response, by definition, there exists $j'$ such that $\mathcal{P}(y_{i^\star} \succ y_{j'}) < 1/2$. Then we have

$$\mathcal{P}_\Psi(\boldsymbol{\delta}_{i^\star}, \boldsymbol{\pi}^\star) = \min_{\boldsymbol{\pi}} \mathcal{P}_\Psi(\boldsymbol{\delta}_{i^\star}, \boldsymbol{\pi}) \leqslant \mathcal{P}_\Psi(\boldsymbol{\delta}_{i^\star}, \boldsymbol{\delta}_{j'}) = \Psi(\mathcal{P}(y_{i^\star} \succ y_{j'})) = M_- . \tag{B.1}$$

However, choosing $i'$ such that $\pi_{i'}^\star > 0$, we have

$$\mathcal{P}_\Psi(\boldsymbol{\delta}_{i^\star}, \boldsymbol{\pi}^\star) = \max_{\boldsymbol{\pi}} \mathcal{P}_\Psi(\boldsymbol{\pi}, \boldsymbol{\pi}^\star) \geqslant \mathcal{P}_\Psi(\boldsymbol{\delta}_{i'}, \boldsymbol{\pi}^\star) = \sum_{i=1}^n \pi_i^\star \Psi(\mathcal{P}(y_{i'} \succ y_i)) > M_- ,$$

which causes a contradiction to (B.1). Hence, we conclude our proof. $\qquad\square$

# C  Proof of Theorem 3.2

*Proof of Theorem 3.2.* First, according to Theorem 3.1, when the Nash solution is Condorcet consistent, we have

$$\begin{cases} \Psi(t) \geqslant \Psi(1/2), 1 \geqslant t \geqslant 1/2 \\ \Psi(t) < \Psi(1/2), 1/2 > t \geqslant 0 \end{cases} . \tag{C.1}$$

In addition, we show that $\Psi(\cdot)$ must satisfy $\Psi(t) + \Psi(1-t) \geqslant 2\Psi(1/2), \forall t \in [0,1]$ for ensuring that the Nash solution is mixed when there is no Condorcet winning response. We consider the case where $n = 4$ and the game with the payoff in Table 3 for any $t_1, t_2 > 1/2$. Notice that if $\Psi(t_1) + \Psi(1-t_1) + \Psi(1/2) \leqslant 3\Psi(1-t_2)$, we have

$$\mathcal{P}_\Psi(\boldsymbol{\delta}_4, \boldsymbol{\pi}) = (\pi_1 + \pi_2 + \pi_3)\Psi(1-t_2) + \pi_4 \Psi(1/2) \Longrightarrow \frac{\boldsymbol{\delta}_1 + \boldsymbol{\delta}_2 + \boldsymbol{\delta}_3}{3} \in \arg\min_{\boldsymbol{\pi}} \mathcal{P}_\Psi(\boldsymbol{\delta}_4, \boldsymbol{\pi}),$$

and

$$\mathcal{P}_\Psi\left(\boldsymbol{\pi}, \frac{\boldsymbol{\delta}_1 + \boldsymbol{\delta}_2 + \boldsymbol{\delta}_3}{3}\right) = (\pi_1 + \pi_2 + \pi_3) \cdot \frac{\Psi(1/2) + \Psi(t_1) + \Psi(1 - t_1)}{3} + \pi_4 \Psi(1 - t_2)$$

$$\implies \boldsymbol{\delta}_4 \in \arg\max_{\boldsymbol{\pi}} \mathcal{P}_\Psi\left(\boldsymbol{\pi}, \frac{\boldsymbol{\delta}_1 + \boldsymbol{\delta}_2 + \boldsymbol{\delta}_3}{3}\right).$$

Therefore, $(\boldsymbol{\delta}_4, (\boldsymbol{\delta}_1 + \boldsymbol{\delta}_2 + \boldsymbol{\delta}_3)/3)$ is a Nash equilibrium, which causes a contradiction to the fact that the Nash solution is mixed. Hence, we have $\Psi(t_1) + \Psi(1 - t_1) + \Psi(1/2) > 3\Psi(1 - t_2)$ for any $t_1, t_2 > 1/2$. Let $t_2 \to 1/2$, we have $\Psi(t) + \Psi(1 - t) \geqslant 2\Psi(1/2)$ for any $t \in [0, 1]$. Hence, combining (C.1), we have shown that the necessary condition for ensuring that the Nash solution is mixed is:

$$\Psi(t) + \Psi(1 - t) \geqslant 2\Psi(1/2)\,, \forall t \in [0, 1] \text{ and } \begin{cases} \Psi(t) \geqslant \Psi(1/2)\,, 1 \geqslant t \geqslant 1/2 \\ \Psi(t) < \Psi(1/2)\,, 1/2 > t \geqslant 0 \end{cases}. \tag{C.2}$$

Next, we prove that the condition (C.2) is also sufficient. Suppose that $(\boldsymbol{\delta}_{i^\star}, \boldsymbol{\pi}^\star)$ is a Nash equilibrium, then we have

$$\mathcal{P}_\Psi(\boldsymbol{\delta}_{i^\star}, \boldsymbol{\pi}^\star) = \max_{\boldsymbol{\pi}} \mathcal{P}_\Psi(\boldsymbol{\pi}, \boldsymbol{\pi}^\star) \geqslant \mathcal{P}_\Psi(\boldsymbol{\pi}^\star, \boldsymbol{\pi}^\star)$$

$$= \sum_{i=1}^n \sum_{j=1}^n \pi_i^\star \pi_j^\star \Psi_{ij} = \frac{1}{2} \sum_{i=1}^n \sum_{j=1}^n \pi_i^\star \pi_j^\star \left(\Psi_{ij} + \Psi_{ji}\right) \geqslant \Psi(1/2)\,.$$

However, notice that for any $j$, we have

$$\mathcal{P}_\Psi(\boldsymbol{\delta}_{i^\star}, \boldsymbol{\pi}^\star) = \min_{\boldsymbol{\pi}} \mathcal{P}_\Psi(\boldsymbol{\delta}_{i^\star}, \boldsymbol{\pi}) \leqslant \mathcal{P}_\Psi(\boldsymbol{\delta}_{i^\star}, \boldsymbol{\delta}_j) = \Psi_{i^\star j}\,.$$

As there is no Condorcet winning response, there must exist $j^\star$ such that $\mathcal{P}(y_{i^\star} \succ y_{j^\star}) < 1/2$, thus $\Psi_{i^\star j^\star} < \Psi(1/2)$. Hence, $\Psi(1/2) \leqslant \mathcal{P}_\Psi(\boldsymbol{\delta}_{i^\star}, \boldsymbol{\pi}^\star) \leqslant \Psi_{i^\star j^\star} < \Psi(1/2)$, which causes a contradiction. Therefore, the Nash solution must be mixed. $\qquad\square$

## D   Proof of Example 4.3

*Proof of Example 4.3.* We prove this conclusion by contradiction. Suppose that the Nash solution is $\boldsymbol{\pi}_1^\star$ that satisfies $\text{supp}(\boldsymbol{\pi}_1^\star) \cap S_1^c \neq \emptyset$, and the Nash equilibrium is $(\boldsymbol{\pi}_1^\star, \boldsymbol{\pi}_2^\star)$.

**Case 1.**   If $\text{supp}(\boldsymbol{\pi}_1^\star) \cap S_1 = \emptyset$, taking $j' \in S_1$, we have

$$\mathcal{P}_\Psi(\boldsymbol{\pi}_1^\star, \boldsymbol{\pi}_2^\star) = \min_{\boldsymbol{\pi}} \mathcal{P}_\Psi(\boldsymbol{\pi}_1^\star, \boldsymbol{\pi}) \leqslant \mathcal{P}_\Psi(\boldsymbol{\pi}_1^\star, \boldsymbol{\delta}_{j'}) = \sum_{i \in S_1^c} \pi_{1,i}^\star \Psi_{ij'} = M_-\,.$$

However, we have

$$\mathcal{P}_\Psi(\boldsymbol{\pi}_1^\star, \boldsymbol{\pi}_2^\star) = \max_{\boldsymbol{\pi}} \mathcal{P}_\Psi(\boldsymbol{\pi}, \boldsymbol{\pi}_2^\star) \geqslant \mathcal{P}_\Psi(\text{Unif}(S_1), \boldsymbol{\pi}_2^\star) = \sum_{i \in S_1} \sum_{j=1}^n \frac{\pi_{2,j}^\star}{|S_1|} \Psi_{ij} > M_-\,,$$

which causes a contradiction.

**Case 2.**   If $\text{supp}(\boldsymbol{\pi}_2^\star) \cap S_1 = \emptyset$ and $\text{supp}(\boldsymbol{\pi}_1^\star) \cap S_1 \neq \emptyset$, taking $i' \in \text{supp}(\boldsymbol{\pi}_1^\star) \cap S_1$, we have

$$\mathcal{P}_\Psi(\boldsymbol{\pi}_1^\star, \boldsymbol{\pi}_2^\star) = \max_{\boldsymbol{\pi}} \mathcal{P}_\Psi(\boldsymbol{\pi}, \boldsymbol{\pi}_2^\star) \geqslant \mathcal{P}_\Psi(\boldsymbol{\delta}_{i'}, \boldsymbol{\pi}_2^\star) = \sum_{j \in S_1^c} \pi_{2,j}^\star \Psi_{i'j} = M_+\,.$$

However, we have

$$\mathcal{P}_\Psi(\boldsymbol{\pi}_1^\star, \boldsymbol{\pi}_2^\star) = \min_{\boldsymbol{\pi}} \mathcal{P}_\Psi(\boldsymbol{\pi}_1^\star, \boldsymbol{\pi}) \leqslant \mathcal{P}_\Psi(\boldsymbol{\pi}_1^\star, \text{Unif}(S_1)) = \sum_{i=1}^n \sum_{j \in S_1} \frac{\pi_{1,i}^\star}{|S_1|} \Psi_{ij} < M_+\,,$$

which cause a contradiction.

**Case 3.** If If $\text{supp}(\boldsymbol{\pi}_2^\star)\cap S_1 \neq \emptyset$ and $\text{supp}(\boldsymbol{\pi}_1^\star)\cap S_1 \neq \emptyset$, taking $i_2^\star \in \text{supp}(\boldsymbol{\pi}_2^\star)\cap S_1$, we consider the following strategy $\boldsymbol{\pi}_1'$:

$$
\begin{cases}
\pi_{1,i}' = 0, & i \in S_1^c \\
\pi_{1,i}' = \pi_{1,i}^\star, & i \in S_1\backslash\{i_2^\star\} \\
\pi_{1,i_2^\star}' = \pi_{1,i_2^\star}^\star + \sum_{i\in S_1^c}\pi_{1,i}^\star, & i = i_2^\star
\end{cases}.
$$

Then we have

$$
\begin{aligned}
\mathcal{P}_\Psi(\boldsymbol{\pi}_1', \boldsymbol{\pi}_2^\star) - \mathcal{P}_\Psi(\boldsymbol{\pi}_1^\star, \boldsymbol{\pi}_2^\star) &= \sum_{i=1}^{n}\sum_{j=1}^{n}\left(\pi_{1,i}'-\pi_{1,i}^\star\right)\pi_{2,j}^\star\Psi_{ij} \\
&= -\sum_{i\in S_1^c}\sum_{j=1}^{n}\pi_{1,i}^\star\pi_{2,j}^\star\Psi_{ij} + \sum_{j=1}^{n}\sum_{i\in S_1^c}\pi_{1,i}^\star\pi_{2,j}^\star\Psi_{i_2^\star j} \qquad \text{(D.1)} \\
&= \sum_{j=1}^{n}\pi_{2,j}^\star\left[\sum_{i\in S_1^c}\pi_{1,i}^\star\left(\Psi_{i_2^\star j}-\Psi_{ij}\right)\right] > 0\,.
\end{aligned}
$$

where the last inequality follows from the following two facts: for any $i \in S_1^c$,

$$
\Psi_{i_2^\star j} - \Psi_{ij} = \begin{cases} M_+ - \Psi_{ij} \geqslant 0, & j \in S_1^c \\ \Psi_{i_2^\star j} - M_- \geqslant 0, & j \in S_1 \end{cases},
$$

and when $j = i_2^\star$,

$$
\pi_{2,i_2^\star}^\star\left[\sum_{i\in S_1^c}\pi_{1,i}^\star\left(\Psi_{i_2^\star i_2^\star}-\Psi_{ii_2^\star}\right)\right] = \pi_{2,i_2^\star}^\star\left(\Psi(1/2)-M_-\right)\sum_{i\in S_1^c}\pi_{1,i}^\star > 0\,.
$$

However, (D.1) causes a contradiction to the fact that $\mathcal{P}_\Psi(\boldsymbol{\pi}_1', \boldsymbol{\pi}_2^\star) \leqslant \max_{\boldsymbol{\pi}}\mathcal{P}_\Psi(\boldsymbol{\pi}, \boldsymbol{\pi}_2^\star) = \mathcal{P}_\Psi(\boldsymbol{\pi}_1^\star, \boldsymbol{\pi}_2^\star)$.

Hence, in summary, it must hold that $\text{supp}(\boldsymbol{\pi}_1^\star)\cap S_1^c = \emptyset$, i.e., $\text{supp}(\boldsymbol{\pi}_1^\star)\subseteq S_1$. $\qquad\square$

# E  Proof of Theorem 4.2

*Proof of Theorem 4.2.* First, we show that the necessary condition for ensuring that problem (1.2) is Smith consistent is:

$$
\Psi(t) + \Psi(1 - t) = 2\Psi(1/2)\,, \forall t \in [0, 1] \quad \text{and} \quad \begin{cases}\Psi(t) \geqslant \Psi(1/2)\,, 1 \geqslant t \geqslant 1/2 \\ \Psi(t) < \Psi(1/2)\,, 0 \leqslant t < 1/2\end{cases}. \qquad \text{(E.1)}
$$

First, Condorcet consistency must hold when Smith consistency holds. According to Theorem 3.1, we have

$$
\begin{cases}\Psi(t) \geqslant \Psi(1/2)\,, 1 \geqslant t \geqslant 1/2 \\ \Psi(t) < \Psi(1/2)\,, 0 \leqslant t < 1/2\end{cases}
$$

Next, we show that when $\Psi(\cdot)$ is continuous at $1/2$, $\Psi(\cdot)$ must satisfy $\Psi(t) + \Psi(1 - t) \geqslant 2\Psi(1/2)$ (Lemma E.1) and $\Psi(t) + \Psi(1-t) \leqslant 2\Psi(1/2)$ (Lemma E.2) for any $t \in [0, 1]$. Therefore, combining the two results together, we obtain the condition (E.1).

**Lemma E.1.** *When $\Psi(\cdot)$ is continuous at $1/2$. Achieving Smith consistency only if*

$$
\Psi(t) + \Psi(1 - t) \geqslant 2\Psi(1/2)\,, \forall t \in [0, 1]\,.
$$

*Proof of Lemma E.1.* We consider the case where $n = 4$ and the game with the payoff in Table 3 for any $t_1, t_2 > 1/2$. Notice that if $\Psi(t_1) + \Psi(1 - t_1) + \Psi(1/2) \leqslant 3\Psi(1 - t_2)$, we have

$$
\mathcal{P}_\Psi(\boldsymbol{\delta}_4, \boldsymbol{\pi}) = (\pi_1 + \pi_2 + \pi_3)\Psi(1 - t_2) + \pi_4\Psi(1/2) \implies \frac{\boldsymbol{\delta}_1 + \boldsymbol{\delta}_2 + \boldsymbol{\delta}_3}{3} \in \arg\min_{\boldsymbol{\pi}}\mathcal{P}_\Psi(\boldsymbol{\delta}_4, \boldsymbol{\pi})\,,
$$

and

$$\mathcal{P}_\Psi\left(\boldsymbol{\pi}, \frac{\boldsymbol{\delta}_1 + \boldsymbol{\delta}_2 + \boldsymbol{\delta}_3}{3}\right) = (\pi_1 + \pi_2 + \pi_3) \cdot \frac{\Psi(1/2) + \Psi(t_1) + \Psi(1 - t_1)}{3} + \pi_4 \Psi(1 - t_2)$$

$$\implies \boldsymbol{\delta}_4 \in \arg\max_{\boldsymbol{\pi}} \mathcal{P}_\Psi\left(\boldsymbol{\pi}, \frac{\boldsymbol{\delta}_1 + \boldsymbol{\delta}_2 + \boldsymbol{\delta}_3}{3}\right).$$

Therefore, $(\boldsymbol{\delta}_4, (\boldsymbol{\delta}_1 + \boldsymbol{\delta}_2 + \boldsymbol{\delta}_3)/3)$ is a Nash equilibrium, which causes a contradiction to the fact that the Nash solution supports on $S_1 := \{y_1, y_2, y_3\}$. Hence, we have $\Psi(t_1) + \Psi(1 - t_1) + \Psi(1/2) > 3\Psi(1 - t_2)$ for any $t_1, t_2 > 1/2$. Let $t_2 \to 1/2$, we have $\Psi(t) + \Psi(1 - t) \geqslant 2\Psi(1/2)$ for any $t \in [0, 1]$. $\qquad\square$

Table 3: Payoff matrix with four responses $\{y_1, y_2, y_3, y_4\}$.

| $\Psi(\mathcal{P}(y \succ y'))$ | $y' = y_1$ | $y' = y_2$ | $y' = y_3$ | $y' = y_4$ |
|---|---|---|---|---|
| $y = y_1$ | $\Psi(1/2)$ | $\Psi(t_1)$ | $\Psi(1 - t_1)$ | $\Psi(t_2)$ |
| $y = y_2$ | $\Psi(1 - t_1)$ | $\Psi(1/2)$ | $\Psi(t_1)$ | $\Psi(t_2)$ |
| $y = y_3$ | $\Psi(t_1)$ | $\Psi(1 - t_1)$ | $\Psi(1/2)$ | $\Psi(t_2)$ |
| $y = y_4$ | $\Psi(1 - t_2)$ | $\Psi(1 - t_2)$ | $\Psi(1 - t_2)$ | $\Psi(1/2)$ |

**Lemma E.2.** *When $\Psi(\cdot)$ is continuous at $1/2$. Achieving Smith consistency only if*

$$\Psi(t) + \Psi(1 - t) \leqslant 2\Psi(1/2), \forall t \in [0, 1].$$

*Proof of Lemma E.2.* We consider the case where $n = 6$ and the game with the payoff in Table 4 for any $t_1, t_2 > 1/2$. Notice that if $\Psi(t_1) + \Psi(1/2) + \Psi(1 - t_1) > 3\Psi(t_2)(\geqslant 3\Psi(1/2) > 3\Psi(1 - t_2))$, there exists positive $\boldsymbol{\mu} = (\mu_1/3, \mu_1/3, \mu_1/3, \mu_2/3, \mu_2/3, \mu_2/3)$ and $\boldsymbol{\mu}' = (\mu_1'/3, \mu_1'/3, \mu_1'/3\mu_2'/3, \mu_2'/3, \mu_2'/3)$ such that $\mu_1 + \mu_2 = \mu_1' + \mu_2' = 1$, and

$$\mu_1 \left[\Psi(1/2) + \Psi(t_1) + \Psi(1 - t_1) - 3\Psi(t_2)\right] = \mu_2 \left[\Psi(1/2) + \Psi(t_1) + \Psi(1 - t_1) - 3\Psi(1 - t_2)\right],$$

$$\mu_1' \left[\Psi(1/2) + \Psi(t_1) + \Psi(1 - t_1) - 3\Psi(1 - t_2)\right] = \mu_2' \left[\Psi(1/2) + \Psi(t_1) + \Psi(1 - t_1) - 3\Psi(t_2)\right].$$

Hence, we have

$$\mu_1(\Psi(1/2) + \Psi(t_1) + \Psi(1 - t_1)) + 3\mu_2\Psi(1 - t_2)$$
$$= \mu_2(\Psi(1/2) + \Psi(t_1) + \Psi(1 - t_1)) + 3\mu_1\Psi(t_2) := 3A,$$
$$\mu_1'(\Psi(1/2) + \Psi(t_1) + \Psi(1 - t_1)) + 3\mu_2'\Psi(t_2)$$
$$= \mu_2'(\Psi(1/2) + \Psi(t_1) + \Psi(1 - t_1)) + 3\mu_1'\Psi(1 - t_2) := 3B.$$

Thus, we have

$$\mathcal{P}_\Psi(\boldsymbol{\pi}, \boldsymbol{\mu}') = (\pi_1 + \pi_2 + \pi_3) \left[\frac{\mu_1'}{3}\left(\Psi(1/2) + \Psi(t_1) + \Psi(1 - t_1)\right) + \mu_2'\Psi(t_2)\right]$$
$$+ (\pi_4 + \pi_5 + \pi_6) \left[\mu_1'\Psi(1 - t_2) + \frac{\mu_2'}{3}\left(\Psi(1/2) + \Psi(t_1) + \Psi(1 - t_1)\right)\right] = B,$$

and

$$\mathcal{P}_\Psi(\boldsymbol{\mu}, \boldsymbol{\pi}) = (\pi_1 + \pi_2 + \pi_3) \left[\frac{\mu_1}{3}\left(\Psi(1/2) + \Psi(t_1) + \Psi(1 - t_1)\right) + \mu_2\Psi(1 - t_2)\right]$$
$$+ (\pi_4 + \pi_5 + \pi_6) \left[\mu_1\Psi(t_2) + \frac{\mu_2}{3}\left(\Psi(1/2) + \Psi(t_1) + \Psi(1 - t_1)\right)\right] = A.$$

Therefore, $\boldsymbol{\mu} \in \arg\max_{\boldsymbol{\pi}} \mathcal{P}_\Psi(\boldsymbol{\pi}, \boldsymbol{\mu}')$, $\boldsymbol{\mu}' \in \arg\min_{\boldsymbol{\pi}} \mathcal{P}_\Psi(\boldsymbol{\mu}, \boldsymbol{\pi})$, which provides that $(\boldsymbol{\mu}, \boldsymbol{\mu}')$ is a Nash equilibrium. However, this causes a contradiction to the fact that the Nash solution supports on $S_1 := \{y_1, y_2, y_3\}$. Thus, it must hold that $\Psi(t_1) + \Psi(1/2) + \Psi(1 - t_1) \leqslant 3\Psi(t_2)$ for any $t_1, t_2 > 1/2$. Let $t_2 \to 1/2$, we obtain $\Psi(t) + \Psi(1 - t) \leqslant 2\Psi(1/2)$ for any $t \in [0, 1]$. $\qquad\square$

Table 4: Payoff matrix with six responses $\{y_1, y_2, y_3, y_4, y_5, y_6\}$.

| $\Psi(\mathcal{P}(y \succ y'))$ | $y' = y_1$ | $y' = y_2$ | $y' = y_3$ | $y' = y_4$ | $y' = y_5$ | $y' = y_6$ |
|---|---|---|---|---|---|---|
| $y = y_1$ | $\Psi(1/2)$ | $\Psi(t_1)$ | $\Psi(1-t_1)$ | $\Psi(t_2)$ | $\Psi(t_2)$ | $\Psi(t_2)$ |
| $y = y_2$ | $\Psi(1-t_1)$ | $\Psi(1/2)$ | $\Psi(t_1)$ | $\Psi(t_2)$ | $\Psi(t_2)$ | $\Psi(t_2)$ |
| $y = y_3$ | $\Psi(t_1)$ | $\Psi(1-t_1)$ | $\Psi(1/2)$ | $\Psi(t_2)$ | $\Psi(t_2)$ | $\Psi(t_2)$ |
| $y = y_4$ | $\Psi(1-t_2)$ | $\Psi(1-t_2)$ | $\Psi(1-t_2)$ | $\Psi(1/2)$ | $\Psi(t_1)$ | $\Psi(1-t_1)$ |
| $y = y_5$ | $\Psi(1-t_2)$ | $\Psi(1-t_2)$ | $\Psi(1-t_2)$ | $\Psi(1-t_1)$ | $\Psi(1/2)$ | $\Psi(t_1)$ |
| $y = y_6$ | $\Psi(1-t_2)$ | $\Psi(1-t_2)$ | $\Psi(1-t_2)$ | $\Psi(t_1)$ | $\Psi(1-t_1)$ | $\Psi(1/2)$ |

Finally, we prove that the condition (E.1) is also sufficient for Smith consistency. Suppose that $(\boldsymbol{\pi}_1^\star, \boldsymbol{\pi}_2^\star)$ is a Nash equilibrium, notice that

$$
\begin{aligned}
\mathcal{P}_\Psi(\boldsymbol{\pi}_1^\star, \boldsymbol{\pi}_2^\star) &= \max_{\boldsymbol{\pi}} \mathcal{P}_\Psi(\boldsymbol{\pi}, \boldsymbol{\pi}_2^\star) \geqslant \mathcal{P}_\Psi(\boldsymbol{\pi}_2^\star, \boldsymbol{\pi}_2^\star) = \Psi(1/2)\,, \\
\mathcal{P}_\Psi(\boldsymbol{\pi}_1^\star, \boldsymbol{\pi}_2^\star) &= \min_{\boldsymbol{\pi}} \mathcal{P}_\Psi(\boldsymbol{\pi}_1^\star, \boldsymbol{\pi}) \leqslant \mathcal{P}_\Psi(\boldsymbol{\pi}_1^\star, \boldsymbol{\pi}_1^\star) = \Psi(1/2)\,,
\end{aligned}
\tag{E.2}
$$

which follows from the following fact: for any $\boldsymbol{\pi}$,

$$
\mathcal{P}_\Psi(\boldsymbol{\pi}, \boldsymbol{\pi}) = \sum_{i=1}^n \sum_{j=1}^n \pi_i \pi_j \Psi_{ij} = \frac{1}{2} \sum_{i=1}^n \sum_{j=1}^n \pi_i \pi_j (\Psi_{ij} + \Psi_{ji}) = \Psi(1/2) \sum_{i=1}^n \sum_{j=1}^n \pi_i \pi_j = \Psi(1/2)\,.
$$

Thus, from (E.2), we have $\mathcal{P}_\Psi(\boldsymbol{\pi}_1^\star, \boldsymbol{\pi}_2^\star) = \Psi(1/2)$. Then we prove $\text{supp}(\boldsymbol{\pi}_1^\star) \subseteq S_1$. Hence, the Nash solution is Smith consistent, i.e., only supports on $S_1$.

**Case 1.** If $\text{supp}(\boldsymbol{\pi}_1^\star) \bigcap S_1 = \emptyset$, taking any $j \in S_1$, we have

$$
\mathcal{P}_\Psi(\boldsymbol{\pi}_1^\star, \boldsymbol{\pi}_2^\star) = \min_{\boldsymbol{\pi}} \mathcal{P}_\Psi(\boldsymbol{\pi}_1^\star, \boldsymbol{\pi}) \leqslant \mathcal{P}_\Psi(\boldsymbol{\pi}_1^\star, \boldsymbol{\delta}_j) = \sum_{i=1}^n \pi_{1,i}^\star \Psi_{ij} = \sum_{i \in S_1^c} \pi_{1,i}^\star \Psi_{ij} < \Psi(1/2)\,,
$$

which causes a contradiction to the fact that $\mathcal{P}_\Psi(\boldsymbol{\pi}_1^\star, \boldsymbol{\pi}_2^\star) = \Psi(1/2)$.

**Case 2.** If $\text{supp}(\boldsymbol{\pi}_1^\star) \bigcap S_1 \neq \emptyset$, and $\text{supp}(\boldsymbol{\pi}_1^\star) \bigcap S_1^c \neq \emptyset$, taking $\widetilde{\boldsymbol{\pi}}_2^\star$ as:

$$
\widetilde{\pi}_{2,j}^\star = \mathbb{1}\{j \in S_1\} \cdot \frac{\pi_{1,j}^\star}{\sum_{j \in S_1} \pi_{1,j}^\star}\,.
$$

Then we have

$$
\begin{aligned}
\mathcal{P}_\Psi(\boldsymbol{\pi}_1^\star, \boldsymbol{\pi}_2^\star) &= \min_{\boldsymbol{\pi}} \mathcal{P}_\Psi(\boldsymbol{\pi}_1^\star, \boldsymbol{\pi}) \leqslant \mathcal{P}_\Psi(\boldsymbol{\pi}_1^\star, \widetilde{\boldsymbol{\pi}}_2^\star) \\
&= \sum_{i \in S_1} \sum_{j \in S_1} \pi_{1,i}^\star \widetilde{\pi}_{2,j}^\star \Psi_{ij} + \sum_{i \in S_1^c} \sum_{j \in S_1} \pi_{1,i}^\star \widetilde{\pi}_{2,j}^\star \Psi_{ij} \\
&< \frac{\sum_{i \in S_1} \sum_{j \in S_1} \pi_{1,i}^\star \pi_{1,j}^\star \Psi_{ij}}{\sum_{j \in S_1} \pi_{1,j}^\star} + \Psi(1/2) \sum_{i \in S_1^c} \sum_{j \in S_1} \pi_{1,i}^\star \widetilde{\pi}_{2,j}^\star \\
&= \Psi(1/2) \sum_{i \in S_1} \pi_{1,i}^\star + \Psi(1/2) \sum_{i \in S_1^c} \pi_{1,i}^\star = \Psi(1/2)\,,
\end{aligned}
\tag{E.3}
$$

which follows from the following fact:

$$
\begin{aligned}
\sum_{i \in S_1} \sum_{j \in S_1} \pi_{1,i}^\star \pi_{1,j}^\star \Psi_{ij} &= \frac{1}{2} \sum_{i \in S_1} \sum_{j \in S_1} \pi_{1,i}^\star \pi_{1,j}^\star (\Psi_{ij} + \Psi_{ji}) \\
&= \Psi(1/2) \sum_{i \in S_1} \sum_{j \in S_1} \pi_{1,i}^\star \pi_{1,j}^\star = \Psi(1/2) \left(\sum_{i \in S_1} \pi_{1,i}^\star\right) \left(\sum_{j \in S_1} \pi_{1,j}^\star\right)
\end{aligned}
$$

However, (E.3) also causes a contradiction to the fact that $\mathcal{P}_\Psi(\boldsymbol{\pi}_1^\star, \boldsymbol{\pi}_2^\star) = \Psi(1/2)$.

Therefore, it must hold that $\text{supp}(\boldsymbol{\pi}_1^\star) \bigcap S_1^c = \emptyset$, i.e., $\text{supp}(\boldsymbol{\pi}_1^\star) \subseteq S_1$. We conclude our proof. $\quad\square$

# F  Proofs of Results in Section 5

## F.1  Proof of Lemma 5.1

*Proof of Lemma 5.1.* Suppose each player has $n$ policies and the payoff matrix is $\{\alpha_{ij}\}_{i=1}^n$. Then,

$$\max_{\boldsymbol{\pi}} \min_{\boldsymbol{\pi}'} \left\{ \sum_{i=1}^n \sum_{j=1}^n \alpha_{ij} \pi_i \pi_j' \right\} = \max_{\boldsymbol{\pi}} \min_{\boldsymbol{\pi}'} \left\{ \sum_{j=1}^n \left( \sum_{i=1}^n \alpha_{ij} \pi_i \right) \pi_j' \right\} = \max_{\boldsymbol{\pi}} \min_{j} \left\{ \sum_{i=1}^n \alpha_{ij} \pi_i \right\}.$$

 Let us reformulated it into a convex optimization problem.

$$\begin{aligned}
\min_{\boldsymbol{\pi}} \quad & \max_{j} \sum_{i=1}^n \alpha_{ij} \pi_i \\
\text{subject to} \quad & -\pi_i \leqslant 0, \quad i = 1, \ldots, n \qquad\qquad (P) \\
& \sum_{i=1}^n \pi_i - 1 = 0
\end{aligned}$$

 Let us further reformulate this problem into the epigraph form by introducing a single variable $t \in \mathbb{R}$:

$$\begin{aligned}
\min_{\boldsymbol{\pi}, t} \quad & t \\
\text{subject to} \quad & \sum_{i=1}^n \alpha_{ij} \pi_i - t \leqslant 0, \quad j = 1, \ldots, n \\
& -\pi_i \leqslant 0, \quad i = 1, \ldots, n \qquad\qquad (P') \\
& \sum_{i=1}^n \pi_i - 1 = 0
\end{aligned}$$

 By introducing the dual variables $\boldsymbol{u}^* \in \mathbb{R}^n$, $\tilde{\boldsymbol{u}}^* \in \mathbb{R}^n$ and $v^* \in \mathbb{R}$, the KKT conditions is:

- stationary condition:

$$\sum_{j=1}^n \alpha_{ij} u_j^* - \tilde{u}_i^* = -v^* \quad i = 1, \cdots, n$$

- complementary slackness:

$$u_j^* \left( \sum_{i=1}^n \pi_i^* \alpha_{ij} - t^* \right) = 0 \quad j = 1, \cdots, n$$

$$\tilde{u}_i^* \pi_i^* = 0 \quad i = 1, \cdots, n$$

- primal feasibility:

$$\sum_{i=1}^n \pi_i^* \alpha_{ij} - t^* \leqslant 0$$

$$\boldsymbol{\pi}^* \geqslant 0$$

$$\sum_{i=1}^n \pi_i^* = 1$$

- dual feasibility:

$$\boldsymbol{u}^* \geqslant 0$$

$$\sum_{i=1}^n u_i^* = 1$$

$$\tilde{\boldsymbol{u}}^* \geqslant 0$$

We can easily see that Slater's condition is satisfied for this problem, so the KKT points are equivalent to primal and dual solutions. Then taking $\boldsymbol{\pi}^* > 0$ into account, we have $\tilde{u}_i^* = 0$ by the second complementary slackness condition, and the above equations can be simplified to the following system of equations:

$$\begin{cases} \boldsymbol{u}^* \geqslant 0 \\ \sum_{i=1}^n u_i^* = 1 \\ \sum_{i=1}^n \pi_i^* \alpha_{ij} - t^* \leqslant 0 & j = 1, \cdots, n \\ u_j^* \left( \sum_{i=1}^n \pi_i^* \alpha_{ij} - t^* \right) = 0 & j = 1, \cdots, n \\ \sum_{j=1}^n \alpha_{ij} u_j^* = -v^* & i = 1, \cdots, n \end{cases}$$

Moreover, notice that

$$0 = \sum_{j=1}^n u_j^* \left( \sum_{i=1}^n \pi_i^* \alpha_{ij} - t^* \right) = \sum_{i=1}^n \pi_i^* \sum_{j=1}^n \alpha_{ij} u_j^* - t^* = -v^* - t^*,$$

thus $v^* = -t^*$. Hence, we conclude our proof. $\qquad\square$

## F.2 Verifying Equation (5.2) and Equation (5.3)

For (5.2), choosing $t^* = -v^* = \sum_{i=1}^n (\pi_i^*)^2$ and $u_i^* = v_i^*$, $\boldsymbol{\pi}^*$ is a Nash solution. For (5.3), choosing $t^* = -v^* = 0$ and $u_j^* = \frac{(\pi_j^*)^{-1}}{\sum_{j=1}^n (\pi_j^*)^{-1}}$, $\boldsymbol{\pi}^*$ is a Nash solution.

## F.3 Proof of Theorem 5.1

We first present a useful lemma (Lemma F.1) that further investigates the KKT conditions (Lemma 5.1) when the payoff matrix induces a unique Nash equilibrium.

**Lemma F.1.** *If a game with the payoff matrix $\{\alpha_{ij}\}_{i,j=1}^n$ has a unique Nash solution $\boldsymbol{\pi}^*$, then for any $j \in [n]$, it must hold $u_j^* > 0$ in the KKT conditions, and*

$$\sum_{i=1}^n \pi_i^* \alpha_{ij} = t^*.$$

*Proof of Lemma F.1.* Suppose that the KKT conditions provide the unique Nash solution $(\boldsymbol{\pi}^*, \boldsymbol{u}^*, t^*)$. Then we define:

$$\mathcal{J}_0 := \left\{ j \in [n] : u_j^* \neq 0 \right\}, \text{ and } \tilde{\mathcal{J}}_0 := \left\{ j \in [n] : u_j^* = 0 \right\},$$

with $\mathcal{J}_0 \cup \tilde{\mathcal{J}}_0 = [n]$. Since $\boldsymbol{u}^* \geqslant 0$ and $\sum u_j^* = 1$, there exists $j \in [n]$, such that $u_j^* \neq 0$, i.e., $\mathcal{J}_0 \neq \emptyset$. Now, we aim to show $\tilde{\mathcal{J}}_0 = \emptyset$. We prove by contradiction. Suppose $\tilde{\mathcal{J}}_0 \neq \emptyset$, taking $j_0 \in \mathcal{J}_0$, we consider two spaces

$$V_1 := \left\{ \boldsymbol{\pi} \in \mathbb{R}^n : \sum_{i=1}^n \pi_i \left( \alpha_{ij} - \alpha_{ij_0} \right) = 0, \ \forall j \in \mathcal{J}_0 \backslash \{j_0\} \right\},$$

$$V_2 := V_1 \bigcap \left\{ \boldsymbol{\pi} \in \mathbb{R}^n : \sum_{i=1}^n \pi_i (\alpha_{ij} - \alpha_{ij_0}) \leqslant 0, \ \forall j \in \tilde{\mathcal{J}}_0 \right\}.$$

Then we claim that $\boldsymbol{\pi}^* \in V_2$ and $\dim(V_2) \geqslant 2$. For the first claim, by the KKT conditions in Lemma 5.1, for any $j \in \mathcal{J}_0$, we obtain

$$\sum_{i=1}^n \pi_i^* \alpha_{ij} = t^* = \sum_{i=1}^n \pi_i^* \alpha_{ij_0},$$

thus $\boldsymbol{\pi}^* \in V_1$. Moreover, again by the KKT conditions, for any $j \in \tilde{\mathcal{J}}_0$, we have

$$\sum_{i=1}^n \pi_i^* \alpha_{ij} \leqslant t^* = \sum_{i=1}^n \pi_i^* \alpha_{ij_0},$$

which shows that $\boldsymbol{\pi}^* \in V_2$. For the second claim, take $\tilde{j}_0 \in \tilde{\mathcal{J}}_0$ and consider

$$V_3 := \left\{ \boldsymbol{\pi} \in \mathbb{R}^n : \sum_{i=1}^n \pi_i \left( \alpha_{ij} - \alpha_{ij_0} \right) = 0, \ \forall j \in [n] \backslash \{j_0, \tilde{j}_0\} \right\},$$

$$V_4 := V_3 \bigcap \left\{ \boldsymbol{\pi} \in \mathbb{R}^n : \sum_{i=1}^n \pi_i (\alpha_{i\tilde{j}_0} - \alpha_{ij_0}) \leqslant 0 \right\}.$$

We can easily see $V_4 \subseteq V_2$. Note that $V_3$ can be regarded as a kernel space of a linear transformation from $\mathbb{R}^n$ to $\mathbb{R}^{n-2}$. By the dimension theorem in linear algebra, we obtain $\dim(V_3) = n - \dim(\text{Im}(A)) \geqslant n - (n-2) = 2$. For any $\boldsymbol{\pi} \in V_3$, it must hold that $\boldsymbol{\pi} \in V_4$ or $-\boldsymbol{\pi} \in V_4$, so $\dim(V_4) = \dim(V_3) \geqslant 2$. Therefore, we have $\dim(V_1) \geqslant \dim(V_2) \geqslant \dim(V_4) \geqslant 2$.

Thus, we can take another $\tilde{\boldsymbol{\pi}}^* \in V_2$ which is linear independent with $\boldsymbol{\pi}^*$. Note that for any $a, b \in \mathbb{R}_+$, $a\boldsymbol{\pi}^* + b\tilde{\boldsymbol{\pi}}^* \in V_2$. Taking large $a \in \mathbb{R}_+$, we have $a\boldsymbol{\pi}^* + b\tilde{\boldsymbol{\pi}}^* \in V_2$ and $a\boldsymbol{\pi}^* + b\tilde{\boldsymbol{\pi}}^* > 0$, since $\boldsymbol{\pi}^* > 0$. Therefore, there exists $a_1 \in \mathbb{R}_+$, such that $\boldsymbol{\pi}_2^* := \frac{a\boldsymbol{\pi}^* + \tilde{\boldsymbol{\pi}}^*}{a_1} \in V_2$ that satisfies $\boldsymbol{\pi}_2^* \neq \boldsymbol{\pi}^*$, $\boldsymbol{\pi}_2^* > 0$, and $\sum_i \pi_{2,i}^* = 1$. Thus, we obtain another Nash equilibrium $(\boldsymbol{\pi}_2^*, \boldsymbol{u}^*, t^*)$, causing contradiction to the uniqueness of Nash solution. Hence, it must hold that $\tilde{\mathcal{J}}_0 = \emptyset$. $\qquad\square$

Next we provide the proof for Theorem 5.1.

*Proof of Theorem 5.1.* Using Lemma F.1, uniqueness requires us to seek solutions that satisfies

$$\sum_{i=1}^n \pi_i^* \alpha_{ij} = t^*$$

for all $j \in [n]$, where $t^*$ is a constant that may depend on $\boldsymbol{\pi}^*$. Consider $n \geqslant 5$, for any four distinct indices $j_1, j_2, k_1, k_2$, we have

$$
\sum_{i \neq j_1, k_1, k_2} \pi_i f \left( \frac{\pi_i}{\pi_{j_1}} \right) + \pi_{k_1} f \left( \frac{\pi_{k_1}}{\pi_{j_1}} \right) + \pi_{k_2} f \left( \frac{\pi_{k_2}}{\pi_{j_1}} \right) + C\pi_{j_1}
$$
$$
= \sum_{i \neq j_2, k_1, k_2} \pi_i f \left( \frac{\pi_i}{\pi_{j_2}} \right) + \pi_{k_1} f \left( \frac{\pi_{k_1}}{\pi_{j_2}} \right) + \pi_{k_2} f \left( \frac{\pi_{k_2}}{\pi_{j_2}} \right) + C\pi_{j_2}
$$
(F.1)

Let us consider the infinitesimal variation $\pi_{k_1} \to \pi_{k_1} + \delta$ and $\pi_{k_2} \to \pi_{k_2} - \delta$, keeping others still. We obtain that

$$
\sum_{i \neq j_1, k_1, k_2} \pi_i f \left( \frac{\pi_i}{\pi_{j_1}} \right) + (\pi_{k_1} + \delta) f \left( \frac{\pi_{k_1} + \delta}{\pi_{j_1}} \right) + (\pi_{k_2} - \delta) f \left( \frac{\pi_{k_2} - \delta}{\pi_{j_1}} \right) + C\pi_{j_1}
$$
$$
= \sum_{i \neq j_2, k_1, k_2} \pi_i f \left( \frac{\pi_i}{\pi_{j_2}} \right) + (\pi_{k_1} + \delta) f \left( \frac{\pi_{k_1} + \delta}{\pi_{j_2}} \right) + (\pi_{k_2} - \delta) f \left( \frac{\pi_{k_2} - \delta}{\pi_{j_2}} \right) + C\pi_{j_2}
$$
(F.2)

Subtracting both sides of (F.1) from (F.2), we obtain that

$$
(\pi_{k_1} + \delta) f \left( \frac{\pi_{k_1} + \delta}{\pi_{j_1}} \right) + (\pi_{k_2} - \delta) f \left( \frac{\pi_{k_2} - \delta}{\pi_{j_1}} \right) - \pi_{k_1} f \left( \frac{\pi_{k_1}}{\pi_{j_1}} \right) - \pi_{k_2} f \left( \frac{\pi_{k_2}}{\pi_{j_1}} \right)
$$
$$
= (\pi_{k_1} + \delta) f \left( \frac{\pi_{k_1} + \delta}{\pi_{j_2}} \right) + (\pi_{k_2} - \delta) f \left( \frac{\pi_{k_2} - \delta}{\pi_{j_2}} \right) - \pi_{k_1} f \left( \frac{\pi_{k_1}}{\pi_{j_2}} \right) - \pi_{k_2} f \left( \frac{\pi_{k_2}}{\pi_{j_2}} \right)
$$
(F.3)

i.e., we have

$$
(\pi_{k_1} + \delta) \left( f \left( \frac{\pi_{k_1} + \delta}{\pi_{j_1}} \right) - f \left( \frac{\pi_{k_1}}{\pi_{j_1}} \right) \right) + \delta f \left( \frac{\pi_{k_1}}{\pi_{j_1}} \right)
$$
$$
+ (\pi_{k_2} - \delta) \left( f \left( \frac{\pi_{k_2} - \delta}{\pi_{j_1}} \right) - f \left( \frac{\pi_{k_2}}{\pi_{j_1}} \right) \right) - \delta f \left( \frac{\pi_{k_2}}{\pi_{j_1}} \right)
$$
$$
= (\pi_{k_1} + \delta) \left( f \left( \frac{\pi_{k_1} + \delta}{\pi_{j_2}} \right) - f \left( \frac{\pi_{k_1}}{\pi_{j_2}} \right) \right) + \delta f \left( \frac{\pi_{k_1}}{\pi_{j_2}} \right)
$$
$$
+ (\pi_{k_2} - \delta) \left( f \left( \frac{\pi_{k_2} - \delta}{\pi_{j_2}} \right) - f \left( \frac{\pi_{k_2}}{\pi_{j_2}} \right) \right) - \delta f \left( \frac{\pi_{k_2}}{\pi_{j_2}} \right).
$$
(F.4)

As $f$ is smooth, using

$$\lim_{\delta \to 0} \frac{f\left(\frac{x+\delta}{\pi_j}\right) - f\left(\frac{x}{\pi_j}\right)}{\delta} = \frac{1}{\pi_j} f'\left(\frac{x}{\pi_j}\right),$$

and taking $\delta \to 0$, we obtain the following identity from (F.4),

$$f\left(\frac{\pi_{k_1}}{\pi_{j_1}}\right) + \frac{\pi_{k_1}}{\pi_{j_1}} f'\left(\frac{\pi_{k_1}}{\pi_{j_1}}\right) - f\left(\frac{\pi_{k_2}}{\pi_{j_1}}\right) - \frac{\pi_{k_2}}{\pi_{j_1}} f'\left(\frac{\pi_{k_2}}{\pi_{j_1}}\right)$$
$$= f\left(\frac{\pi_{k_1}}{\pi_{j_2}}\right) + \frac{\pi_{k_1}}{\pi_{j_2}} f'\left(\frac{\pi_{k_1}}{\pi_{j_2}}\right) - f\left(\frac{\pi_{k_2}}{\pi_{j_2}}\right) - \frac{\pi_{k_2}}{\pi_{j_2}} f'\left(\frac{\pi_{k_2}}{\pi_{j_2}}\right). \tag{F.5}$$

Thus, we obtain that

$$f\left(\frac{\pi_{k_1}}{\pi_{j_1}}\right) + \frac{\pi_{k_1}}{\pi_{j_1}} f'\left(\frac{\pi_{k_1}}{\pi_{j_1}}\right) - f\left(\frac{\pi_{k_1}}{\pi_{j_2}}\right) - \frac{\pi_{k_1}}{\pi_{j_2}} f'\left(\frac{\pi_{k_1}}{\pi_{j_2}}\right)$$
$$= f\left(\frac{\pi_{k_2}}{\pi_{j_1}}\right) + \frac{\pi_{k_2}}{\pi_{j_1}} f'\left(\frac{\pi_{k_2}}{\pi_{j_1}}\right) - f\left(\frac{\pi_{k_2}}{\pi_{j_2}}\right) - \frac{\pi_{k_2}}{\pi_{j_2}} f'\left(\frac{\pi_{k_2}}{\pi_{j_2}}\right). \tag{F.6}$$

Since (F.6) holds for any $\pi > 0$, given any $\pi_{j_1} \neq \pi_{j_2}$, for any $x_1, x_2 \in (0, 1 - \pi_{j_1} - \pi_{j_2})$, we have

$$f\left(\frac{x_1}{\pi_{j_1}}\right) + \frac{x_1}{\pi_{j_1}} f'\left(\frac{x_1}{\pi_{j_1}}\right) - f\left(\frac{x_1}{\pi_{j_2}}\right) - \frac{x_1}{\pi_{j_2}} f'\left(\frac{x_1}{\pi_{j_2}}\right)$$
$$= f\left(\frac{x_2}{\pi_{j_1}}\right) + \frac{x_2}{\pi_{j_1}} f'\left(\frac{x_2}{\pi_{j_1}}\right) - f\left(\frac{x_2}{\pi_{j_2}}\right) - \frac{x_2}{\pi_{j_2}} f'\left(\frac{x_2}{\pi_{j_2}}\right),$$

which induces the following for any $x \in (0, 1 - \pi_{j_1} - \pi_{j_2})$,

$$f\left(\frac{x}{\pi_{j_1}}\right) + \frac{x}{\pi_{j_1}} f'\left(\frac{x}{\pi_{j_1}}\right) - f\left(\frac{x}{\pi_{j_2}}\right) - \frac{x}{\pi_{j_2}} f'\left(\frac{x}{\pi_{j_2}}\right) = C(\pi_{j_1}, \pi_{j_2}). \tag{F.7}$$

i.e., we have

$$f\left(\frac{x}{\pi_{j_1}}\right) + \frac{x}{\pi_{j_1}} f'\left(\frac{x}{\pi_{j_1}}\right) = C(\pi_{j_1}, \pi_{j_2}) + f\left(\frac{x}{\pi_{j_2}}\right) + \frac{x}{\pi_{j_2}} f'\left(\frac{x}{\pi_{j_2}}\right). \tag{F.8}$$

Without any loss of generality, we assume $\pi_{j_1} < \pi_{j_2}$, then we obtain

$$f\left(\frac{x}{\pi_{j_1}}\right) + \frac{x}{\pi_{j_1}} f'\left(\frac{x}{\pi_{j_1}}\right)$$
$$= C(\pi_{j_1}, \pi_{j_2}) + f\left(\frac{x}{\pi_{j_2}}\right) + \frac{x}{\pi_{j_2}} f'\left(\frac{x}{\pi_{j_2}}\right)$$
$$= 2C(\pi_{j_1}, \pi_{j_2}) + f\left(\frac{\pi_{j_1} x}{\pi_{j_2}^2}\right) + \frac{\pi_{j_1} x}{\pi_{j_2}^2} f'\left(\frac{\pi_{j_1} x}{\pi_{j_2}^2}\right)$$
$$= \cdots\cdots$$
$$= nC(\pi_{j_1}, \pi_{j_2}) + f\left(\frac{\pi_{j_1}^{n-1} x}{\pi_{j_2}^n}\right) + \frac{\pi_{j_1}^{n-1} x}{\pi_{j_2}^n} f'\left(\frac{\pi_{j_1}^{n-1} x}{\pi_{j_2}^n}\right)$$
$$= \cdots\cdots .$$

Taking limit, it must hold $C(\pi_{j_1}, \pi_{j_2}) = 0$, i.e., we have

$$f\left(\frac{x}{\pi_{j_1}}\right) + \frac{x}{\pi_{j_1}} f'\left(\frac{x}{\pi_{j_1}}\right) = f\left(\frac{x}{\pi_{j_2}}\right) + \frac{x}{\pi_{j_2}} f'\left(\frac{x}{\pi_{j_2}}\right). \tag{F.9}$$

Since (F.9) holds for any $\pi > 0$, for any $x_1, x_2 \in \mathbb{R}_+$, we have

$$f(x_1) + x_1 f'(x_1) = f'(x_2) + x_2 f'(x_2),$$

thus, for any $x \in \mathbb{R}_+$, we have

$$f(x) + x f'(x) = C_1. \tag{F.10}$$

Solving (F.10), we obtain that

$$f(x) = \frac{C_2}{x} + C_3 \, .$$

Then we obtain that

$$\sum_{i=1}^{n} \pi_i \alpha_{ij} = C\pi_j + \sum_{i \neq j} \pi_i \left( \frac{C_2 \pi_j}{\pi_i} + C_3 \right) = C_3 + (C + (n-1)C_2 - C_3)\pi_j \, ,$$

yielding $C_3 = C + (n-1)C_2$, and

$$f(x) = C + C_2 \left( \frac{1}{x} + n - 1 \right) ,$$

which is contradictory to our assumptions. $\qquad\square$

