# OpenReview forum: "Fundamental Limits of Game-Theoretic LLM Alignment: Smith Consistency and Preference Matching"
_NeurIPS.cc/2025/Conference — Submitted to NeurIPS 2025_

### Official Review · Reviewer_rouN · 2025-06-25

**Clarity:** 3
**Significance:** 2
**Originality:** 2
**Rating:** 4
**Confidence:** 2

**Summary:**

The paper analyzes the NLHF framework for aligning LLMs to human preference, by investigating payoff functions that map raw human preferences. The paper builds upon prior work and shows exact conditions that the payoff functions must satisfy in order for the Nash solution to the NLHF problem to converge to the Concordet winner if it exists, or to stochastically select multiple possible optimal responses, if the Concordet winner does not exist, in order to reflect diversity in human preference. Similar conditions are found to identify the Smith set. Further, it is proven that it is impossible under NLHF to learn a payoff matrix that leads to preference matching.

**Questions:**

Can you elaborate on lines 201-202 that continuous mappings are commonly encountered in practice? It would seem that this is not the case when there are few data points for example.

In line 259, do you mean to say "human preferences ~~does not~~ satisfy"? Is this a typo?

Why does the symmetry of the payoff matrix 5.2 make it difficult to interpret? I do not understand your comment on line 326.

You show in your impossibility result that preference matching is impossible. Could you think of how it is possible to achieve this approximately, with some relaxation? You could mention this in the discussions.

*Minor typographical errors:*
Line 13: highlight -> highlights,
Line 77: is insensitive -> are insensitive,
Line 87: reveal -> reveals,
Line 139: response -> responses,
Line 140: denote -> denotes,
Line 211: investigate -> investigates,
Line 223: capture -> captures,
Line 225: similar property -> a similar property,
Line 248: mappings that is -> mappings that are,
Line 259: does not -> do not,
Line 300: do not accounts for -> do not account for,
Line 301: under BTL model -> under the BTL model,
Line 318: write KKT in full the first time you use it

**Ethical Concerns:**

["NO or VERY MINOR ethics concerns only"]

**Final Justification:**

My problem with this paper was that it did not provide enough discussion on its practical use while being purely theoretical, but the authors' responses in the rebuttal phase towards this were convincing.

**Limitations:**

The no-tie assumption is pretty significant. Even an individual labeler, nevermind in aggregate, may wish to indicate a tie on a specific pair of responses especially if they are very similar. I believe this assumption should be highlighted in the limitations section.

Further, the work is purely theoretical. A potential avenue for future work, which should be mentioned in the discussions, is to perform simulations with different payoff functions to observe their effect on (the ease of) identifying the Concordet winner, Smith set, or capturing diversity in human preference.

**Paper Formatting Concerns:**

No major formatting issues were noticed.

**Quality:**

3

**Strengths And Weaknesses:**

Strengths: \
The paper conducts a thorough theoretical analysis of the NLHF framework under generic payoff functions. Given the closeness with prior work, the contributions are clearly presented and distinguished. Additionally, each theoretical result is well explained and easy to follow.

Weaknesses: \
The main contribution is the generalization of prior results on NLHF to general payoff functions. However, I do not understand why one would consider a payoff function other than the identity one. For this reason, I cannot recommend the paper for acceptance without a strong argument, or an example, or a simulation, of alternative payoff functions that can be useful in practice. I will increase my score if you expand on the practical implications of your results.

---

> ### Author Rebuttal · Authors · 2025-07-31
>
> We thank Reviewer rouN for the comments and questions. Below we provide our responses.
> ___
> **Q1.** *However, I do not understand why one would consider a payoff function other than the identity one.*
>
>
> **A.**  Thank you for the question. First and foremost, we would like to clarify that the main form of $\Psi$ we consider—emphasized in the introduction (lines 70–71)—is:
>
> $$
> \mathcal{P}_\theta(y \succ y') = \Psi(\mathcal{P}(y \succ y')),
> $$
>
> where $\mathcal{P}_\theta$ denotes the empirical preference model used in practice.
>
> One key practical motivation for studying non-identity transformations $\Psi$ is to analyze **how empirical preference models used in real-world NLHF implementations** (which may be distorted or biased) affect the outcome of alignment. Our theoretical framework allows us to examine the implications of using such transformed preferences and to identify conditions under which desirable alignment properties are preserved or violated.
>
>  In practice, we do not have access to the ideal ground-truth preference model $\mathcal{P}$. Instead, we rely on finite data and limited optimization to train an empirical model $\mathcal{P}_\theta$, which inevitably introduces bias and noise. As a result, although the identity map of a ground true preference model  in our framework satisfies properties such as Condorcet consistency, Smith consistency, and diversity considerations in theory, it remains unclear whether these properties hold when the empirical model replaces the ideal one. To address this, we introduce a general mapping \Psi that represents the transformation from the ideal preference model to the empirical one:
>
> Specifically, if we treat $\Psi$ as a random function, then $\Psi(\mathcal{P}(y \succ y’))$ serves as a stochastic estimate of the ideal preference, denoted by $\mathcal{P}_\theta(y \succ y’)$, thereby capturing the practical uncertainties inherent in NLHF.
>
> In addition, our analysis also applies to scenarios where an additional mapping is introduced into the NLHF framework. Therefore, our results provide a general theoretical framework for analyzing such extensions.
>
>
>
> ___
> **Q2**. *For this reason, I cannot recommend the paper for acceptance without a strong argument, or an example, or a simulation, of alternative payoff functions that can be useful in practice.*
>
>
> **A**. This framework enables us to analyze practical NLHF and derive several key insights:
>
> - To achieve Condorcet consistency, the requirement on the empirical preference model $\mathcal{P} _\theta$ is relatively mild. It only needs to preserve correct pairwise comparisons. Specifically, if the true preference model satisfies $\mathcal{P}(y \succ y') > 1/2$—meaning that more than half prefer response $y$ over $y'$—then the empirical model should also satisfy $\mathcal{P} _\theta(y \succ  y') > 1/2$. Importantly, the exact proportion of preference is not required to be highly accurate. This indicates that some degree of bias in $\mathcal{P} _\theta$ is acceptable, as long as the direction of pairwise preferences is preserved, highlighting the robustness of practical NLHF.
>
>
> - When there is noise in the empirical preference model, it becomes difficult for practical NLHF to satisfy all the desired properties because when $\mathcal{P}(y \succ y’)$ is close to 1/2, even small amounts of noise can easily lead the empirical model to make incorrect pairwise comparisons. In such cases, although the empirical preference model may still be a good approximation of the ideal one—with only minor noise—it can nevertheless cause NLHF to fail in satisfying all these properties. This makes us consider several modifications to solve this issue:
>
>   - If the pair $(y, y^\prime)$ corresponds to a preference value close to $1/2$, we exclude it from the alignment process.
>
>   - We can incorporate regularization terms into the training process of the preference model to encourage it to output higher values when the true preference exceeds $1/2$, and lower values when it falls below $1/2$.
>
>   - If we assume that the empirical preference model closely approximates the true preference and only contains noise, we can use Bayesian inference to estimate the probability that the true preference exceeds $1/2$, given an empirical value near $1/2$. Based on this estimate, we can adjust the preference values to increase the likelihood of making correct pairwise comparisons.
>
> Exploring practically useful modifications and developing efficient alignment methods that incorporate them is left for future work.
>
> Moreover, our general framework introduces additional payoff functions for game-theoretic LLM alignment, enabling the achievement of several desirable properties.
> ___
> **Q3**. *Could you elaborate on lines 201–202, where you state that continuous mappings are commonly encountered in practice? This may not hold true, for example, when there are few data points.*
>
> **A.** The continuity referred to here is a property of the mapping $\Psi$ itself and is independent of the number of data points. Even with limited data, the mapping can be defined to be continuous.
> ___
> **Q4**. *In line 259, do you mean to say "human preferences does not satisfy"? Is this a typo?*
>
> **A.** Thank you so much for this valuable question! This is not a typo, but it might be confusing here. We have revised the sentences as follows in our manuscript:  When human preference does not satisfy BTL model, this formula can be viewed as a natural generalization of standard RLHF. And our results imply that this generalized formula satisfies Smith consistency.
> ___
> **Q5**. *Why does the symmetry of the payoff matrix 5.2 make it difficult to interpret? I do not understand your comment on line 326.*
>
> **A.** Here, symmetry refers to the condition $\alpha_{ij} = \alpha_{ji}$. However, we interpret $\alpha_{ij}$ as the payoff when response $i$ beats response $j$, and $\alpha_{ji}$ as the payoff when response $j$ beats response $i$. From this perspective, increasing $\alpha_{ij}$ should correspond to decreasing $\alpha_{ji}$. For instance, they are typically expected to satisfy an anti-symmetry condition such as $\alpha_{ij} + \alpha_{ji} = 1$. Therefore, we find this construction unintuitive and difficult to interpret.
> ___
> **Q6**. *You show in your impossibility result that preference matching is impossible. Could you think of how it is possible to achieve this approximately, with some relaxation? You could mention this in the discussions.*
>
> **A.** Incorporating regularization terms into the training objective may be a promising direction for enabling preference matching. Exploring specific strategies to overcome the current impossibility results is left for future work.
> ___
>
> **Q7**. *Minor typographical errors … The no-tie assumption is pretty significant. Even an individual labeler, nevermind in aggregate, may wish to indicate a tie on a specific pair of responses especially if they are very similar. I believe this assumption should be highlighted in the limitations section. Further, the work is purely theoretical. A potential avenue for future work, which should be mentioned in the discussions, is to perform simulations with different payoff functions to observe their effect on (the ease of) identifying the Condorcet winner, Smith set, or capturing diversity in human preference.*
>
> **A.** Thank you very much for pointing out the typos and highlighting important limitations. We have corrected the typos and added a more detailed discussion of the no-tie assumption, along with additional future directions, to the limitations section of our manuscript.

---

> > ### Comment · Reviewer_rouN · 2025-08-05
> >
> > Thank you for your detailed response addressing all of my questions and concerns. I am satisfied with the given motivations for the paper regarding the $\Psi$ mapping and the insights that follow from the results, and recommend highlighting them in the manuscript since multiple reviewers questioned the paper's practical use. I will increase my score.

---

> > > ### Author Response · Authors · 2025-08-07
> > >
> > > Thank you very much for your response. We have revised our manuscript to clarify the motivations behind the $\Psi$ mapping and the associated insights. We appreciate your thoughtful feedback and the increase in score.

---

### Official Review · Reviewer_7tZ6 · 2025-06-30

**Clarity:** 3
**Significance:** 2
**Originality:** 3
**Rating:** 4
**Confidence:** 2

**Summary:**

This paper discusses several topics about the Nash Learning from Human Feedback (NLHF) framework. It starts with the introduction and formulation of a generalised NLHF framework with a mapping $\Psi$ over preference probabilities. Then it discusses conditions $\Psi$ has to satisfy in order to make the generalised NLHF problem Condorcet consistent and Smith consistent. Besides, it presents an impossibility result in preference matching.

**Questions:**

Please find the weaknesses points above. The most important missing part of this paper is (1) the motivation of $\Psi$ and (2) the empirical relevance of findings presented. I personally find this paper interesting but am unsure about how its usefulness.

**Ethical Concerns:**

["NO or VERY MINOR ethics concerns only"]

**Final Justification:**

I am satisfied with the rebuttal. Specifically, the authors have provided (1) the motivation of generalising NLHF, and (2) clarification on the preference matching part. I would like to modify the score accordingly.

**Limitations:**

Yes, the paper discussed some of its limitations.

**Paper Formatting Concerns:**

No paper formatting concerns.

**Quality:**

2

**Strengths And Weaknesses:**

**Strengths**:

- The paper is very-well written. The introduction and preliminary sections provided enough knowledge to understand section 3-5. The theoretical results are also clearly presented in a fluent flow. In general, this paper is easy to understand and pleasant to read.
- Studying the properties of NLHF based on a generalised framework itself does improve our understanding. Nevertheless, its relevancy is unclear in the current manuscript.

**Weaknesses**:

- The motivation for generalising NLHF is unclear. What problem is this paper trying to solve? Why do we want to generalise NLHF with a mapping $\Psi$? Does this provide better understanding about performance guarantees in practice? The paper mentioned $\Psi$PO, but the motivation of introducing $\Psi$ is very clearly articulated in that paper as a generalisation of RLHF and DPO.
- The analysis for preference matching in section 5 seems detached from the two consistencies introduced in the earlier sections of the paper.
- Lack of empirical study and practical relevance. While this submission is clearly a theoretical paper, its theoretical claims should be demonstrated by empirical results. These will be helpful to show, for example, benefits we get in practice if the two consistencies in section 3 and 4 are guaranteed. Alternatively, the paper should at least practically demonstrate what we are risking if such consistencies are missing. Otherwise, it is hard to tell if the findings in this paper is relevant.

---

> ### Author Rebuttal · Authors · 2025-07-31
>
> We thank Reviewer 7tZ6 for the comments and questions. Below we provide our responses.
> ___
> **Q1.** *The motivation for generalising NLHF is unclear. What problem is this paper trying to solve? Why do we want to generalise NLHF with a mapping? Does this provide better understanding about performance guarantees in practice? The paper mentioned PO, but the motivation of introducing  is very clearly articulated in that paper as a generalisation of RLHF and DPO.*
>
> **A.**  Thank you for the question. First and foremost, we would like to clarify that the main form of $\Psi$ we consider—emphasized in the introduction (lines 70–71)—is:
>
> $$
> \mathcal{P}_\theta(y \succ y') = \Psi(\mathcal{P}(y \succ y')),
> $$
>
> where $\mathcal{P}_\theta$ denotes the empirical preference model used in practice.
>
> One key practical motivation for studying non-identity transformations $\Psi$ is to analyze **how empirical preference models used in real-world NLHF implementations** (which may be distorted or biased) affect the outcome of alignment. Our theoretical framework allows us to examine the implications of using such transformed preferences and to identify conditions under which desirable alignment properties are preserved or violated.
>
>  In practice, we do not have access to the ideal ground-truth preference model $\mathcal{P}$. Instead, we rely on finite data and limited optimization to train an empirical model $\mathcal{P}_\theta$, which inevitably introduces bias and noise. As a result, although the identity map of a ground true preference model  in our framework satisfies properties such as Condorcet consistency, Smith consistency, and diversity considerations in theory, it remains unclear whether these properties hold when the empirical model replaces the ideal one. To address this, we introduce a general mapping \Psi that represents the transformation from the ideal preference model to the empirical one:
>
> Specifically, if we treat $\Psi$ as a random function, then $\Psi(\mathcal{P}(y \succ y’))$ serves as a stochastic estimate of the ideal preference, denoted by $\mathcal{P}_\theta(y \succ y’)$, thereby capturing the practical uncertainties inherent in NLHF.
>
> In addition, our analysis also applies to scenarios where an additional mapping is introduced into the NLHF framework. Therefore, our results provide a general theoretical framework for analyzing such extensions.
>
> ---
>
> **Q2.** *The analysis for preference matching in Section 5 seems detached from the two consistencies introduced in the earlier sections of the paper.*
>
> **A:** Thank you for the question. Let us clarify the connection between preference matching and the two consistency properties discussed earlier in the paper. Our analysis is organized around two hierarchical goals in human preference alignment:
>
> **Primary Goal of Alignment: Aligning with majority preference:**
>
> To improve the overall performance of LLMs, it is critical to align the model with the *most preferred* response—specifically, the **Condorcet winner** (if it exists), or more generally, the **Smith set**. This is why we focus on whether NLHF satisfies **Condorcet consistency** and **Smith consistency** in Sections 3 and 4.
>
> **Secondary Goal of Alignment: Preserving minority preferences:**
>
> After aligning with the majority-preferred responses, a natural secondary goal is to ensure that **minority preferences are not disregarded**. This is where **preference matching** becomes important: it captures the idea that an aligned LLM should reflect the full distribution of preferences—including those of minority groups—rather than collapsing entirely to majority views.
>
> In Section 5, we analyze preference matching as a formal property that supports this secondary goal. Although this analysis is distinct from Condorcet and Smith consistency, it complements them by addressing fairness and diversity in generation after majority alignment is achieved.
>
> Below, we further clarify the connection between preference matching and preserving minority preference. First, we would like to clarify the distinction between two concepts of preference matching: the **original** concept, which is independent of a reward function, and the **equivalent** concept discussed in our paper.
>
> **Original concept of preference matching**:
>
> A policy $\pi$ is said to satisfy *preference matching* if
>
> $$\frac{\pi(y)}{\pi(y')} = \frac{\mathcal{P}(y \mid y')}{\mathcal{P}(y' \mid y)}.$$
>
> This definition does not rely on an explicit reward model. Its intuitive meaning is clear: for instance, if 70% of people prefer $y$ over $y'$, and 30% prefer $y'$ over $y$, then ideally, the LLM should generate $y$ and $y'$ with probabilities in a 7:3 ratio.
>
> However, not all human preferences can be represented by a policy satisfying the above condition. In fact, it has been shown in previous studies that this is possible if and only if human preferences can be modeled by a BTL model. In that case, the original concept of preference matching is equivalent to the following formulation.
>
> **Equivalent concept of preference matching**:
>
> A policy $\pi$ is said to satisfy *preference matching* if
>
> $$\pi(y) = \frac{\exp(r(y))}{\sum_{y'} \exp(r(y'))},$$
>
> where $r(y)$ is a latent utility score associated with response $y$.
>
> We agree that introducing the equivalent concept directly—without first discussing the original one—can be misleading. We have revised this part in the paper to improve clarity.
>
> In our paper, we reflect this hierarchy of objectives. In Sections 3 and 4, we first address the **primary goal**—aligning the policy with the Condorcet winner and Smith set. Then, in Section 5, we consider a **secondary yet important goal**—ensuring alignment with minority preferences through preference matching.
>
> ---
> **Q3.** *Lack of empirical study and practical relevance. While this submission is clearly a theoretical paper, its theoretical claims should be demonstrated by empirical results. These would help show, for example, the benefits of satisfying the two consistency properties in Sections 3 and 4. Alternatively, the paper should at least demonstrate, in practice, what is at risk if these consistencies are violated. Otherwise, it is hard to assess the practical relevance of the findings.*
>
> **A:**  Thank you for the thoughtful suggestion. We would like to emphasize that this line of work—studying the axiomatic foundations of aligning LLMs with human feedback—is inherently theoretical in nature. Several papers in this area have focused purely on theoretical contributions and have been accepted at NeurIPS. For example:
>
> - [1] Noothigattu et al. (2020) propose axioms for learning from pairwise comparisons without empirical validation.
> - [2] Ge et al. (2024) develop axioms for alignment from human feedback, again focusing on theoretical properties.
>
> We appreciate the reviewer’s concern about practical relevance. However, our paper focuses on analyzing the theoretical properties of existing alignment approaches, rather than proposing a new algorithm. As such, empirical experiments are not essential for validating our contributions.
>
> Given this context, we believe that pure theoretical work of this nature aligns with the NeurIPS standard.
>
> **References**
> [1] Noothigattu, R., Peters, D., & Procaccia, A. D. (2020). Axioms for learning from pairwise comparisons. *NeurIPS*, 33, 17745–17754.
> [2] Ge, L., Halpern, D., Micha, E., Procaccia, A. D., Shapira, I., Vorobeychik, Y., & Wu, J. (2024). Axioms for AI alignment from human feedback. *NeurIPS*, 37, 80439–80465.
>
> ---
> **Q4.** *The empirical relevance of findings presented.*
>
> **A**. This framework enables us to analyze practical NLHF and derive several key insights:
>
> - To achieve Condorcet consistency, the requirement on the empirical preference model $\mathcal{P} _\theta$ is relatively mild. It only needs to preserve correct pairwise comparisons. Specifically, if the true preference model satisfies $\mathcal{P}(y \succ y') > 1/2$—meaning that more than half prefer response $y$ over $y'$—then the empirical model should also satisfy $\mathcal{P} _\theta(y \succ  y') > 1/2$. Importantly, the exact proportion of preference is not required to be highly accurate. This indicates that some degree of bias in $\mathcal{P} _\theta$ is acceptable, as long as the direction of pairwise preferences is preserved, highlighting the robustness of practical NLHF.
>
>
> - When there is noise in the empirical preference model, it becomes difficult for practical NLHF to satisfy all the desired properties because when $\mathcal{P}(y \succ y’)$ is close to 1/2, even small amounts of noise can easily lead the empirical model to make incorrect pairwise comparisons. In such cases, although the empirical preference model may still be a good approximation of the ideal one—with only minor noise—it can nevertheless cause NLHF to fail in satisfying all these properties. This makes us consider several modifications to solve this issue:
>
>   - If the pair $(y, y^\prime)$ corresponds to a preference value close to $1/2$, we exclude it from the alignment process.
>   - We can incorporate regularization terms into the training process of the preference model to encourage it to output higher values when the true preference exceeds $1/2$, and lower values when it falls below $1/2$.
>   - If we assume that the empirical preference model closely approximates the true preference and only contains noise, we can use Bayesian inference to estimate the probability that the true preference exceeds $1/2$, given an empirical value near $1/2$. Based on this estimate, we can adjust the preference values to increase the likelihood of making correct pairwise comparisons.
>
>  Exploring practically useful modifications and developing efficient alignment methods that incorporate them is left for future work.
>
> Moreover, our general framework introduces additional payoff functions for game-theoretic LLM alignment, enabling the achievement of several desirable properties.

---

> > ### Comment · Reviewer_7tZ6 · 2025-08-08
> >
> > Thank you for the reply. I am satisfied with the rebuttal and will update my score accordingly.

---

### Official Review · Reviewer_g65k · 2025-07-02

**Clarity:** 3
**Significance:** 2
**Originality:** 2
**Rating:** 3
**Confidence:** 4

**Summary:**

This paper investigates what kinds of payoff matrices in game-theoretic LLM alignment can satisfy desirable alignment properties, including Condorcet consistency and Smith consistency. The authors also provide a theoretical analysis showing that no payoff matrix can ensure the Nash solution is unique and equals to the preference-matching policy under the Bradley-Terry (BT) model.

**Questions:**

A key motivation behind game-theoretic LLM alignment is to move beyond the Bradley-Terry model assumption and consider general preferences. Given this, why is it necessary to focus on the preference-matching policy and design a payoff matrix such that the Nash solution corresponds to it?

**Ethical Concerns:**

["NO or VERY MINOR ethics concerns only"]

**Final Justification:**

My concern about the practical implications of the introduced Condorcet and Smith consistency remains. Since the identity mapping naturally satisfies both properties, it remains unclear how these consistency notions can meaningfully guide the development of alignment algorithms, particularly under settings with a noisy preference oracle.

**Quality:**

2

**Strengths And Weaknesses:**

Strengths:

a. Exploring notions such as Condorcet consistency and Smith consistency in the context of LLM alignment is an interesting and novel direction.

b. The presentation is clear and easy to follow.

Weaknesses:

a. The discussion on Condorcet and Smith consistency is interesting, but I do not see its broader implications. The widely used identity mapping already satisfies both properties, and it's unclear how this discussion provides practical insights or inspiration.

b. In game-theoretic LLM alignment, there is no explicit reward function $r(x, y)$, so it is unclear why the Nash solution should correspond to a preference-matching policy. Moreover, I am not convinced that learning a preference-matching policy should be the primary goal in this setting. More explanation is needed to clarify this point.

I find the overall contribution of the paper to be limited. The core challenge in RLHF lies in designing practical and effective algorithms for LLM post-training. However, the theoretical analysis presented in the paper offers limited insight into how such algorithms could be developed or improved.

---

> ### Author Rebuttal · Authors · 2025-07-31
>
> We thank Reviewer g65k for the comments and questions. Below we provide our responses.
> ___
> **Q1.** *The discussion on Condorcet and Smith consistency is interesting, but I do not see its broader implications. The widely used identity mapping already satisfies both properties, and it's unclear how this discussion provides practical insights or inspiration.*
>
> **A.**  Thank you for the question. First and foremost, we would like to clarify that the main form of $\Psi$ we consider—emphasized in the introduction (lines 70–71)—is:
>
> $$
> \mathcal{P}_\theta(y \succ y') = \Psi(\mathcal{P}(y \succ y')),
> $$
>
> where $\mathcal{P}_\theta$ denotes the empirical preference model used in practice.
>
> One key practical motivation for studying non-identity transformations $\Psi$ is to analyze **how empirical preference models used in real-world NLHF implementations** (which may be distorted or biased) affect the outcome of alignment. Our theoretical framework allows us to examine the implications of using such transformed preferences and to identify conditions under which desirable alignment properties are preserved or violated.
>
>  In practice, we do not have access to the ideal ground-truth preference model $\mathcal{P}$. Instead, we rely on finite data and limited optimization to train an empirical model $\mathcal{P}_\theta$, which inevitably introduces bias and noise. As a result, although the identity map of a ground true preference model  in our framework satisfies properties such as Condorcet consistency, Smith consistency, and diversity considerations in theory, it remains unclear whether these properties hold when the empirical model replaces the ideal one. To address this, we introduce a general mapping \Psi that represents the transformation from the ideal preference model to the empirical one:
>
> Specifically, if we treat $\Psi$ as a random function, then $\Psi(\mathcal{P}(y \succ y’))$ serves as a stochastic estimate of the ideal preference, denoted by $\mathcal{P}_\theta(y \succ y’)$, thereby capturing the practical uncertainties inherent in NLHF.
>
> In addition, our analysis also applies to scenarios where an additional mapping is introduced into the NLHF framework. Therefore, our results provide a general theoretical framework for analyzing such extensions.
>
>
> ___
> **Q2.1.** *In game-theoretic LLM alignment, there is no explicit reward function, so it is unclear why the Nash solution should correspond to a preference-matching policy.*
>
> **A:** Thank you for the question. We would like to clarify the distinction between two concepts of preference matching: the **original** concept, which is independent of a reward function, and the **equivalent** concept discussed in our paper.
>
> **Original concept of preference matching**:
> A policy $\pi$ is said to satisfy *preference matching* if
>
> $$\frac{\pi(y)}{\pi(y')} = \frac{P(y \mid y')}{P(y' \mid y)}.$$
>
> This definition does not rely on an explicit reward model. Its intuitive meaning is clear: for instance, if 70% of people prefer $y$ over $y'$, and 30% prefer $y'$ over $y$, then ideally, the LLM should generate $y$ and $y'$ with probabilities in a 7:3 ratio.
>
> However, not all human preferences can be represented by a policy satisfying the above condition. In fact, it has been shown in previous studies that this is possible if and only if human preferences can be modeled by a BTL model. In that case, the original concept of preference matching is equivalent to the following formulation.
>
> **Equivalent concept of preference matching**:
> A policy $\pi$ is said to satisfy *preference matching* if
>
> $$\pi(y) = \frac{\exp(r(y))}{\sum_{y'} \exp(r(y'))},$$
>
> where $r(y)$ is a latent utility score associated with response $y$.
>
> We agree that introducing the equivalent concept directly—without first discussing the original one—can be misleading. We have revised this part in the paper to improve clarity.
> ___
> **Q 2.2.** *Moreover, I am not convinced that learning a preference-matching policy should be the primary goal in this setting. More explanation is needed to clarify this point.*
>
> **A:**  Thank you for the question. We agree that the primary goal of training LLMs is to improve performance. However, other aspects are also critical in alignment. Since LLMs are generative models, maintaining **generation diversity** is an important consideration. An ideal alignment algorithm should achieve both high performance and diversity, ensuring that the model does not collapse onto majority preferences alone.
>
> Preference matching plays a key role in this regard: it preserves generative diversity and protects minority opinions, which are often suppressed under majority-focused objectives.
>
> In our paper, we reflect this hierarchy of objectives. In Sections 3 and 4, we first address the **primary goal**—aligning the policy with the Condorcet winner and Smith set. Then, in Section 5, we consider a **secondary yet important goal**—ensuring alignment with minority preferences through preference matching.
> ___
> **Comment 1.** *I find the overall contribution of the paper to be limited. The core challenge in RLHF lies in designing practical and effective algorithms for LLM post-training. However, the theoretical analysis presented in the paper offers limited insight into how such algorithms could be developed or improved.*
>
> **A:** Thank you for the comment. First, although our paper is primarily theoretical, it also offers **practical insights**.  This framework enables us to analyze practical NLHF and derive several key insights:
>
> - To achieve Condorcet consistency, the requirement on the empirical preference model $\mathcal{P} _\theta$ is relatively mild. It only needs to preserve correct pairwise comparisons. Specifically, if the true preference model satisfies $\mathcal{P}(y \succ y') > 1/2$—meaning that more than half prefer response $y$ over $y'$—then the empirical model should also satisfy $\mathcal{P} _\theta(y \succ  y') > 1/2$. Importantly, the exact proportion of preference is not required to be highly accurate. This indicates that some degree of bias in $\mathcal{P} _\theta$ is acceptable, as long as the direction of pairwise preferences is preserved, highlighting the robustness of practical NLHF.
>
>
> - When there is noise in the empirical preference model, it becomes difficult for practical NLHF to satisfy all the desired properties because when $\mathcal{P}(y \succ y’)$ is close to 1/2, even small amounts of noise can easily lead the empirical model to make incorrect pairwise comparisons. In such cases, although the empirical preference model may still be a good approximation of the ideal one—with only minor noise—it can nevertheless cause NLHF to fail in satisfying all these properties. This makes us consider several modifications to solve this issue:
>
>   - If the pair $(y, y^\prime)$ corresponds to a preference value close to $1/2$, we exclude it from the alignment process.
>
>   - We can incorporate regularization terms into the training process of the preference model to encourage it to output higher values when the true preference exceeds $1/2$, and lower values when it falls below $1/2$.
>
>   - If we assume that the empirical preference model closely approximates the true preference and only contains noise, we can use Bayesian inference to estimate the probability that the true preference exceeds $1/2$, given an empirical value near $1/2$. Based on this estimate, we can adjust the preference values to increase the likelihood of making correct pairwise comparisons.
>
> Exploring practically useful modifications and developing efficient alignment methods that incorporate them is left for future work.
>
> Moreover, our general framework introduces additional payoff functions for game-theoretic LLM alignment, enabling the achievement of several desirable properties.
>
>
> In addition, from a technical perspective, we introduce **novel proof techniques** that go beyond prior work. Specifically, we develop tools to analyze general *non-symmetric* games directly—without relying on the symmetry assumptions that underlie previous results in NLHF. We believe these techniques can serve as a foundation for future theoretical and algorithmic advancements in preference-based alignment.
> ___
> **Q3.** *A key motivation behind game-theoretic LLM alignment is to move beyond the Bradley-Terry model assumption and consider general preferences. Given this, why is it necessary to focus on the preference-matching policy and design a payoff matrix such that the Nash solution corresponds to it?*
>
> **A:**  Thank you for the question. We respond to this point in part in our response to Question 2, but we are happy to elaborate further here.
>
> First, we emphasize that NLHF is designed to handle **general preferences**, not limited to any specific parametric form such as the BTL model. The axiomatic approach studies the **desirable properties** that a solution should satisfy *when certain conditions hold*—this is common in the theoretical analysis of social choice and decision-making systems.
>
> For example:
> - In general preferences, a Condorcet winner may not exist. NLHF operates on general preferences, but when a Condorcet winner *does* exist, it is desirable for NLHF to return it. This property is referred to as Condorcet consistency.
> - Similarly, in general preferences, preference matching policy may not exist. **NLHF operates on general preferences without such policy.** However, when the preference-matching policy exists (i.e., can be represented by a BTL model), it is desirable for NLHF to return the **preference-matching policy**.
>
> This is why we study (though do not exclusively focus on) the performance of NLHF under the BTL assumption. Doing so allows us to characterize what alignment behavior should look like in idealized, structured settings—while still preserving the generality of the overall framework.

---

> > ### Comment · Reviewer_g65k · 2025-08-06
> >
> > Thank you for the rebuttal. I have some follow-up questions:
> >
> > 1. I am unclear about the relationship between the noisy estimated model and the introduced Condorcet and Smith consistency. According to Theorems 3.2 and 4.2, these consistency notions seem to pertain only to the mapping function itself. I could not find results in the paper analyzing the effect of noise in the preference model. For example, are functions that satisfy Condorcet or Smith consistency more robust to noise in the preference labels?
> >
> > 2. My understanding of Section 5 is that it shows there is no mapping function for which the NE of the game is always the preference-matching policy. What practical insights or considerations should we draw from this result? Should we still aim for the preference-matching policy in practice?

---

> > > ### Author Response · Authors · 2025-08-08
> > >
> > > We thank Reviewer g65k for the follow-up questions.
> > >
> > > ---
> > >
> > > **Q1.** *I am unclear about the relationship between the noisy estimated model and the introduced Condorcet and Smith consistency. ... For example, are functions that satisfy Condorcet or Smith consistency more robust to noise in the preference labels?*
> > >
> > > **A.** Thank you for the question. To clarify a potential misunderstanding: our paper does not use the function $\Psi$ to handle noise in the preference labels or preference matrix. Instead, the (stochastic) function $\Psi$ is used to model or represent the noise in these preferences. Informally speaking, the function $\Psi$ is the noise in the preference labels.
> > >
> > > According to Theorems 3.2 and 4.2, if the function $\Psi$ satisfies certain inequalities—i.e., if the noise levels are bounded in a specific way—then the max-min problem can still guarantee Condorcet and Smith consistency.
> > >
> > > Below, we provide a more detailed explanation.
> > >
> > > Let $\mathcal{P}(y \succ y')$ denote the ground-truth preference model, and let $\mathcal{P}_\theta(y \succ y')$ denote the practical (estimated) preference model. The latter is a noisy approximation of the former, where the noise may arise from label noise or other factors.
> > >
> > > We use a (stochastic) function $\Psi$ to model this noise:
> > >
> > > $$
> > > \mathcal{P}_\theta(y \succ y') = \Psi\left(\mathcal{P}(y \succ y')\right).
> > > $$
> > >
> > > For example, consider the following additive noise model:
> > >
> > > $$
> > > \mathcal{P}_\theta(y \succ y') = \mathcal{P}(y \succ y') + \varepsilon\left(\mathcal{P}(y \succ y')\right),
> > > $$
> > >
> > > where $\varepsilon\left(\mathcal{P}(y \succ y')\right)$ captures the noise. In this case, the function $\Psi$ is given by
> > >
> > > $$
> > > \Psi(\cdot) = \cdot + \varepsilon(\cdot).
> > > $$
> > > Theorem 3.2 shows that, if the noise term satisfy
> > >
> > > $$ \varepsilon(\mathcal{P}(y\succ y^\prime))\geq 1/2-\mathcal{P}(y\succ y^\prime),\text{if}, \mathcal{P}(y\succ y^\prime)\geq1/2,$$
> > > and
> > > $$\varepsilon(\mathcal{P}(y\succ y^\prime))< 1/2 - \mathcal{P}(y\succ y^\prime), \text{if},, \mathcal{P}(y\succ y^\prime)<1/2,$$
> > >
> > > The max-min problem can guarantee Condorcet consistency.
> > >
> > > When the true preference value is far from 1/2, the requirements on the noise term for achieving Condorcet consistency are relatively mild. However, when the true preference value is close to $1/2$, the noise must be more carefully controlled to maintain Condorcet consistency. Furthermore, to ensure Smith consistency (Theorem 4.2), the noise term must satisfy the above conditions and also fulfill the requirement $\varepsilon(\mathcal{P}(y \succ y^\prime)) + \varepsilon(\mathcal{P}(y^\prime \succ y)) = 0$.
> > >
> > > ---
> > >
> > > **Q2.** My understanding of Section 5 is that it shows there is no mapping function for which the NE of the game is always the preference-matching policy. What practical insights or considerations should we draw from this result? Should we still aim for the preference-matching policy in practice?
> > >
> > > **A.** Thank you for the question. Our short answers are as follows:
> > >
> > > - For the first question, when a practitioner or developer aims to achieve a preference-matching policy, designing payoff matrices alone is insufficient.
> > >
> > > - For the second question, if the goal is to develop models that promote fairness or diversity, then one should indeed aim for a preference-matching policy.
> > >
> > > Below, we provide a more detailed explanation, beginning with the second question and then returning to the first.
> > >
> > > In this work, we investigate two types of alignment properties for LLMs. The primary goal is to preserve majority preferences, formalized through Condorcet and Smith consistency. The secondary goal is to safeguard minority preferences, which we capture through preference matching. This ensures that the aligned LLM does not collapse entirely to the majority view.
> > >
> > > Therefore, if a developer is only concerned with aligning to majority preferences, ensuring Condorcet or Smith consistency may be sufficient. However, if a **practitioner cares about model diversity or fairness**—i.e., also values the preservation of minority preferences—then preference matching becomes essential, and the design **should explicitly aim for a preference-matching policy**.
> > >
> > > Now, we return to the first question. In this work, we show that game-theoretic LLM alignment has a fundamental limitation: it cannot achieve preference matching through any smooth transformation of the preference model. This leads to an important **practical insight—designing payoff matrices alone is insufficient. To achieve preference matching, additional techniques, such as applying regularization to the policy, could be considered.**
> > >
> > > For instance, prior studies have shown that standard RLHF fails to achieve preference matching, but introducing an entropy regularization term can help mitigate this issue. Analogously, incorporating specific regularization terms into game-theoretic LLM alignment may offer a promising direction for overcoming this limitation, which we leave for future work.

---

### Official Review · Reviewer_ua5U · 2025-07-05

**Clarity:** 3
**Significance:** 3
**Originality:** 3
**Rating:** 5
**Confidence:** 3

**Summary:**

Nash Learning from Human Feedback  is a game-theoretic framework for aligning LLMs with human preferences by modeling learning as a two-player zero-sum game. The paper investigates how different payoff functions derived from pairwise preferences impact alignment, establishing conditions for Condorcet consistency, diversity through mixed strategies, and Smith consistency. A key theoretical result shows that preference matching, which ensures a unique Nash equilibrium that aligns with a target policy, is impossible even under standard preference models such as Bradley-Terry-Luce (BTL). This reveals a fundamental limitation in game-theoretic LLM alignment, suggesting that while NLHF provides a structured approach, perfect alignment with human preferences cannot be guaranteed through Nash equilibria alone.

**Questions:**

Questions:

1. Can your results be extended beyond NLHF? For example, can your results be extended to GRPO?

2. Can you give specific examples of \Psi(t) and run the experiments to verify your theory？

3. Recent work [1] has shown that preference embedding, an approach that embeds responses into a latent space to capture intricate preference structures can handle preference cycles (e.g.,  the Condorcet cycle discussed in your paper). Do your theoretical results, particularly the impossibility of preference matching, have any implications for this work? For instance, does your analysis suggest inherent limitations even in the preference embedding model, or could your analysis be adapted to accommodate such generalizations?

[1] Beyond Bradley-Terry Models: A General Preference Model for Language Model Alignment, ICML 2025.

**Ethical Concerns:**

["NO or VERY MINOR ethics concerns only"]

**Final Justification:**

This is a theoretical paper that provides some practical guides. The theoretical results are sound. The authors have addressed all my questions.

**Limitations:**

This is pure theoretical work and its immediate practical impact is unclear.

**Quality:**

3

**Strengths And Weaknesses:**

Strengths:

1. The paper provides rigorous theoretical analysis, establishing sufficient and necessary conditions for Condorcet consistency and Smith consistency, which significantly deepens the understanding of Nash Learning from Human Feedback (NLHF). These results offer valuable insights into the robustness and limitations of game-theoretic alignment approaches.

2. The impossibility result on preference matching is another theoretical contribution, which reveals fundamental challenges in aligning LLMs with human preferences via Nash equilibria, even under standard assumptions like the BTL model. This finding could guide future research toward alternative alignment methods.

Weaknesses:

1. While the theoretical contributions are strong, the work lacks empirical validation or practical implementation guidelines. Since NLHF is not yet widely adopted in real-world LLM alignment (unlike reinforcement learning from human feedback, RLHF), its practical value is limited.

---

> ### Author Rebuttal · Authors · 2025-07-31
>
> We thank Reviewer ua5U for the comments and questions. Below we provide our responses.
> ___
> **Weakness 1.** *This work is intended to be a fully theoretical paper that can guide... However, our theoretical results indeed provide practical implementation guidelines…*
>
> **A.** This framework enables us to analyze practical NLHF and derive several key insights:
>
> - To achieve Condorcet consistency, the requirement on the empirical preference model $\mathcal{P} _\theta$ is relatively mild. It only needs to preserve correct pairwise comparisons. Specifically, if the true preference model satisfies $\mathcal{P}(y \succ y') > 1/2$—meaning that more than half prefer response $y$ over $y'$—then the empirical model should also satisfy $\mathcal{P} _\theta(y \succ  y') > 1/2$. Importantly, the exact proportion of preference is not required to be highly accurate. This indicates that some degree of bias in $\mathcal{P} _\theta$ is acceptable, as long as the direction of pairwise preferences is preserved, highlighting the robustness of practical NLHF.
>
>
> - When there is noise in the empirical preference model, it becomes difficult for practical NLHF to satisfy all the desired properties because when $\mathcal{P}(y \succ y’)$ is close to 1/2, even small amounts of noise can easily lead the empirical model to make incorrect pairwise comparisons. In such cases, although the empirical preference model may still be a good approximation of the ideal one—with only minor noise—it can nevertheless cause NLHF to fail in satisfying all these properties. This makes us consider several modifications to solve this issue:
>
>   - If the pair $(y, y^\prime)$ corresponds to a preference value close to $1/2$, we exclude it from the alignment process.
>   - We can incorporate regularization terms into the training process of the preference model to encourage it to output higher values when the true preference exceeds $1/2$, and lower values when it falls below $1/2$.
>
>   - If we assume that the empirical preference model closely approximates the true preference and only contains noise, we can use Bayesian inference to estimate the probability that the true preference exceeds $1/2$, given an empirical value near $1/2$. Based on this estimate, we can adjust the preference values to increase the likelihood of making correct pairwise comparisons.
>
> Exploring practically useful modifications and developing efficient alignment methods that incorporate them is left for future work.
>
> Moreover, our general framework introduces additional payoff functions for game-theoretic LLM alignment, enabling the achievement of several desirable properties.
>
> ___
> **Q1.** *Can your results be extended beyond NLHF? For example, can your results be extended to GRPO?*
>
> **A:** In our analysis, we consider a general preference-based game-theoretic alignment method, where NLHF is a special example with \Psi being the identity. However, as we understood, although the reward can be interpreted as preference in Group Relative Policy Optimization (GRPO,[2]) as it is either 0 or 1, GRPO is inherently not a game-theoretic alignment method. Therefore, our results cannot be directly extended to GRPO.
> ___
> **Q2.** *Can you give specific examples of \Psi(t) and run the experiments to verify your theory*
>
> **A:** As noted above, \Psi being the identity is the traditional NLHF. Another special example of \Psi that is worthy to note is \Psi(t)=log(t/(1-t)), which is a natural game-theoretical extension of traditional RLHF, as we pointed out in line 258-259 in the paper. As all our theorems are proven with mathematical rigor, there is no need to run experiments to verify the theorems. Nevertheless, we recognize that testing the empirical performance of different \Psi on language models is important, but it is beyond the scope of this theoretical paper.
> ___
> **Q3.** *Recent work [1] has shown that preference embedding, an approach that embeds responses into a latent space to capture intricate preference structures can handle preference cycles (e.g., the Condorcet cycle discussed in your paper). Do your theoretical results, particularly the impossibility of preference matching, have any implications for this work? For instance, does your analysis suggest inherent limitations even in the preference embedding model, or could your analysis be adapted to accommodate such generalizations?*
>
> **A:** Thank you for your insightful question. In [1], the authors proposed the General Preference embedding model (GPM) for modeling preference, which can then be used in preference-based LLM alignment, including non game-theoretical approaches (like \PsiPO or the GPO proposed in this paper), and the game-theoretical approach discussed in our paper.
>
> Theorem 4.2 in our paper, as discussed in line 266-269, suggests that the preference model used should be skew-symmetry to guarantee Smith consistency and previous preference models like PairRM are not skew-symmetry. The GPM in [1] is designed to be skew-symmetry, therefore our result implies that using GPM is better than PairPM in preserving Smith consistency.
>
> Theorem 5.1 (the impossibility of preference matching) applies to any game-theoretical alignment method, regardless of the preference model used. Therefore if we use the GPM model and a **game-theoretical** alignment method, this limitation will persist.
> We have added the above discussions into our paper. Thank you again for bringing [1] to our attention.
>
> **References**
>
> [1] Beyond Bradley-Terry Models: A General Preference Model for Language Model Alignment, ICML 2025.
> [2] DeepSeekMath: Pushing the Limits of Mathematical Reasoning in Open Language Models.

---

> > ### Comment · Reviewer_ua5U · 2025-08-09
> >
> > Thank you for addressing my questions. I will maintain my score

---

### Decision · Program_Chairs · 2025-09-17

**Decision:**

Reject

**Comment:**

The paper provides a theoretical analysis of game-theoretic LLM alignment, establishing conditions for Condorcet and Smith consistency and demonstrating impossibility results regarding preference matching. While the theoretical contributions are technically solid and the manuscript is clearly presented, several reviewers raised concerns about its limited practical significance and absence of empirical validation. Although empirical experiments are not strictly required for NeurIPS submissions, the practical applications of this research remain unclear. Specifically, reviewers questioned what practical problems could directly benefit from this theory and whether new practical algorithms might be inspired by these results. In addition to empirical evaluation, these concerns could be effectively addressed by including a more detailed discussion connecting the theory to real-world scenarios or by providing intuitive toy examples motivated by realistic contexts. We encourage the authors to strengthen their manuscript in these directions. With such improvements, we believe the paper would make a substantial contribution to the ML community.